# Neuronal detection triggers systemic digestive shutdown in response to adverse food sources in *Caenorhabditis elegans*

Yating Liu[†], Guojing Tian[†], Ziyi Wang, Junkang Zheng, Huimin Liu, Sucheng Zhu, Zhao Shan*, Bin Qi*

Southwest United Graduate School, Yunnan Key Laboratory of Cell Metabolism and Diseases, State Key Laboratory of Conservation and Utilization of Bio-resources in Yunnan, Center for Life Sciences, School of Life Sciences, Yunnan University, Kunming, China

## eLife Assessment

This **important** study investigates how signals from the nervous system can influence the response to different food sources. To demonstrate the role of specific neuronal and intestinal regulators in sensing food quality and modulating digestion, the authors present evidence through a combination of genetic screening, RNA-seq analysis, and functional studies. These findings shed light on an adaptive strategy to integrate food perception with physiological responses, with a mix of **solid** and **convincing** evidence supporting the work.

**\*For correspondence:**
shanzhaolab@163.com (ZS);
qb@ynu.edu.cn (BQ)

[†]These authors contributed equally to this work

**Competing interest:** The authors declare that no competing interests exist.

**Abstract** The ability to sense and adapt to adverse food conditions is essential for survival across species, but the detailed mechanisms of neuron-digestive crosstalk in food sensing and adaptation remain poorly understood. Here, we identify a novel mechanism by which *Caenorhabditis elegans* detect unfavorable food sources through neurons and initiate a systemic response to shut down digestion, thus safeguarding against potential harm. Specifically, we demonstrate that NSY-1, expressed in AWC neurons, detects *Staphylococcus saprophyticus* (SS) as an unfavorable food source, prompting the animal to avoid and halt digestion of SS. Upon detection, the animals activate the AWC[OFF] neural circuit, leading to a systemic digestive shutdown, which is mediated by NSY-1-dependent STR-130. Additionally, NSY-1 mutation triggers the production of insulin peptides, including INS-23, which interact with the DAF-2 receptor to modulate SS digestion and affect the expression of intestinal BCF-1. These findings uncover a crucial survival strategy through neuron-digestive crosstalk, where the NSY-1 pathway in AWC neurons orchestrates food evaluation and initiates digestive shutdown to adapt effectively to harmful food sources.

## Introduction

Food is a source of essential nutrients and also poses a risk of lethal toxins and pathogens. Animals, including humans, must respond to various food sources to ensure survival. The ability to detect and adapt to adverse food conditions is crucial for the survival of many species. Various sensory mechanisms have evolved to monitor food quality by detecting beneficial and harmful substances, including olfactory (*Fiala, 2007*; *McLachlan et al., 2022*; *Sengupta et al., 1996*), gustatory (*Avery et al., 2021*; *Hukema et al., 2006*; *Scott, 2018*), and gut chemosensory systems (*Bargmann, 2006*). Food allergies

**eLife digest** Eating is essential for survival – but not all food is safe. Spoiled or toxic meals can cause illness, so animals must distinguish good food from harmful food. While the brain helps animals smell and taste, it is less clear how the nervous system communicates with the digestive system to prevent harm.

The tiny worm *Caenorhabditis elegans* (*C. elegans*) is a powerful model organism in biology because it has a simple nervous system and a transparent body. Living in soil and feeding on bacteria, the worm encounters both harmless and harmful species. One such bacterium, *Staphylococcus saprophyticus*, is toxic to *C. elegans*. Previous work showed that worms can avoid poor-quality food, but the mechanisms behind this behavior were unknown.

Liu et al. investigated how *C. elegans* detects and responds to dangerous food by exposing the worms to *S. Saprophyticus* for one to four days and by using a combination of genetic and imaging approaches to study the activity of neurons. With this approach, the team identified a pair of neurons in the worm's head, called AWC neurons, as key "taste sentinels."

A protein located in these neurons, NSY-1, enabled the worms to recognize *S. saprophyticus* as a threat. This detection triggered a neural circuit (the AWC$^{OFF}$ state), sending a body-wide signal that shut down the digestive system. Without this protective mechanism governed by the *nsy-1* gene, worms continued to digest the toxic bacteria and had a shortened lifespan.

Further experiments revealed that these neural signals also regulated hormone-like peptides and gut-specific genes, fine-tuning digestive activity. Thus, NSY-1 functions as a molecular sensor that links the nervous system to the gut, forming a direct communication line that helps the animal avoid harm.

These findings reveal a fundamental survival mechanism that may represent an ancient system shared across animals, including humans. Understanding this brain–gut crosstalk in worms could provide insights into how the human nervous system defends against foodborne pathogens and toxins and may also illuminate the biological basis of some digestive disorders.

However, further research is needed to determine whether similar signaling pathways exist in mammals. Identifying equivalent molecules in humans could open new avenues for understanding and treating digestive disorders and food-related illnesses.

---

serve as a biological food quality control system, offering protection and benefits by promoting food avoidance behavior (*Florsheim et al., 2021*). Research by *Plum et al., 2023* and *Florsheim et al., 2023* provides evidence that the immune system's allergic response communicates with the brain in mice, leading to food avoidance. This avoidance behavior acts as a defense strategy, reducing the risk of exposure to harmful substances, including allergens.

The digestive system functions by transporting food through the gastrointestinal (GI) tract, where it is broken down into molecules that can be absorbed and utilized by the body's cells. Thus, shutdown of digestion may serve as a mechanism for eliminating indigestible or harmful substances, acting as a protective system in animals to avoid adverse food. Despite this, the interaction between neuronal food detection and intestinal digestion, particularly in assessing and adapting to harmful food, remains inadequately understood.

The free-living nematode *Caenorhabditis elegans* thrives in organic-rich environments where it encounters a variety of microorganisms as food (*Félix and Braendle, 2010*; *Samuel et al., 2016*; *Schulenburg and Félix, 2017*). *C. elegans* has evolved mechanisms to sense bacterial presence and food quality, which influence its feeding behaviors and digestive processes to adapt to its environment. Previous research has identified heat-killed *Escherichia coli* as low-quality food, which the nematode avoids using its food-quality evaluation systems, such as the FAD-ATP-TORC1-ELT-2 pathway (*Qi et al., 2017*) and the UPR$^{ER}$–immunity pathway (*Liu et al., 2024*). Additionally, we have found that certain bacteria, like *Staphylococcus saprophyticus* (SS), are classified as inedible, leading to shutting down digestion and stunted growth in the nematodes (*Geng et al., 2022*). However, we still need to determine whether SS represents harmful food for *C. elegans* and how the nematode senses SS and shuts down digestion to reject it. It is hypothesized that *C. elegans* may detect or assess inedible food, such as SS, and subsequently halt digestion as a survival strategy. This suggests that the cooperation between food sensing and digestive systems could

form a systemic food quality control mechanism in animals, aimed at minimizing the adverse effects of harmful food.

In this study, we developed a model in *C. elegans* to investigate responses to harmful food and explored how the nematodes sense and avoid ingesting such food by shutting down their digestive systems. We identified a food quality control mechanism involving communication between neurons and the digestive system. This mechanism functions as a defense strategy to minimize the adverse effects of harmful food environments on the animals.

## Results

### Shutting down digestion as a protective mechanism for survival in larval *C. elegans*

Previous studies have shown that *C. elegans* cannot digest SS, which prevents their development (*Figure 1A*). In natural environments, *C. elegans* rely on various bacteria for nutrition and growth. However, SS is not a viable food source for *C. elegans*. Larval arrest, such as the dauer stage, serves as an adaptive mechanism for survival under unfavorable conditions, including limited food availability or extreme temperatures (*Baugh, 2013*; *Baugh and Hu, 2020*). We hypothesize that *C. elegans* can sense or evaluate inedible food, such as SS, and subsequently shut down their digestion to arrest development as a protective survival strategy (*Figure 1B*).

Our previous research demonstrated that *C. elegans* can assess and avoid low-quality food, like heat-killed *E. coli*, to adapt to nutrient-deficient conditions (*Liu et al., 2024*; *Qi et al., 2017*). To determine if *C. elegans* also detect SS as an unfavorable food source, we conducted two behavioral assays: food dwelling/avoidance and food choice (*Qi et al., 2017*). In the food dwelling/avoidance assay, larval-stage animals exhibited strong discrimination against SS compared to the standard food, OP50 (*Figure 1C*). In the food choice assay, the animals preferred OP50 over low-quality food such as heat-killed *E. coli* (*Figure 1—figure supplement 1A*) or SS (*Figure 1D*). However, they could not distinguish between heat-killed *E. coli* and SS (*Figure 1—figure supplement 1B*). These results suggest that SS acts as an unfavorable food that *C. elegans* can detect and avoid.

In response to starvation, L1 larvae can enter a state of developmental arrest, pausing their growth to survive (*Baugh and Sternberg, 2006*). To test whether *C. elegans* shut down digestion of SS as a protective strategy upon sensing unfavorable food, we performed a survival assay on L1 larvae fed with SS. We found that larvae unable to digest SS still survived under SS feeding conditions (*Figure 1—figure supplement 1C*), similar to larvae under L1 starvation. This suggests that shutting down digestion may be a protective mechanism in larvae under SS feeding conditions.

Previously, we observed that activating larval digestion with heat-killed *E. coli* or *E. coli* cell wall peptidoglycan (PGN) enabled the digestion of SS as food (*Hao et al., 2024*). Additionally, when animals reached the L2 stage by feeding normal OP50 diet, they could utilize SS as a food source to support growth (*Figure 1—figure supplement 1D*). These findings suggest that once digestion is activated (via *E. coli* components or L2-stage maturation), worms gain the capacity to process SS as a viable food source, abolishing SS-induced growth impairment (*Hao et al., 2024*; *Figure 1—figure supplement 1D*).

If SS is indeed an unfavorable or toxic food for *C. elegans*, digesting it could result in physiological defects. We measured the lifespan of L4 stage animals fed with SS or OP50. We found that SS consumption shortened their lifespan (*Figure 1E and F*), indicating a cost associated with digesting unfavorable or toxic food.

In conclusion, our data suggest that larval-stage *C. elegans* can sense and evaluate SS as an unfavorable food source, leading to the shutdown of digestion to avoid consumption, thereby protecting them and allowing adaptation to an unfavorable food environment.

### *C. elegans* sense SS and shut down digestion through NSY-1

We speculated that key factors in *C. elegans* are involved in sensing SS and shutting down its digestion. If these factors are mutated, the animals would fail to detect SS as an unfavorable food source and would utilize it (*Figure 2—figure supplement 1A*). To identify these factors, we conducted an unbiased forward genetic screen to find mutant animals that cannot sense SS, thereby allowing its digestion and supporting growth. One of the mutant alleles identified, *ylf6*, could digest SS and

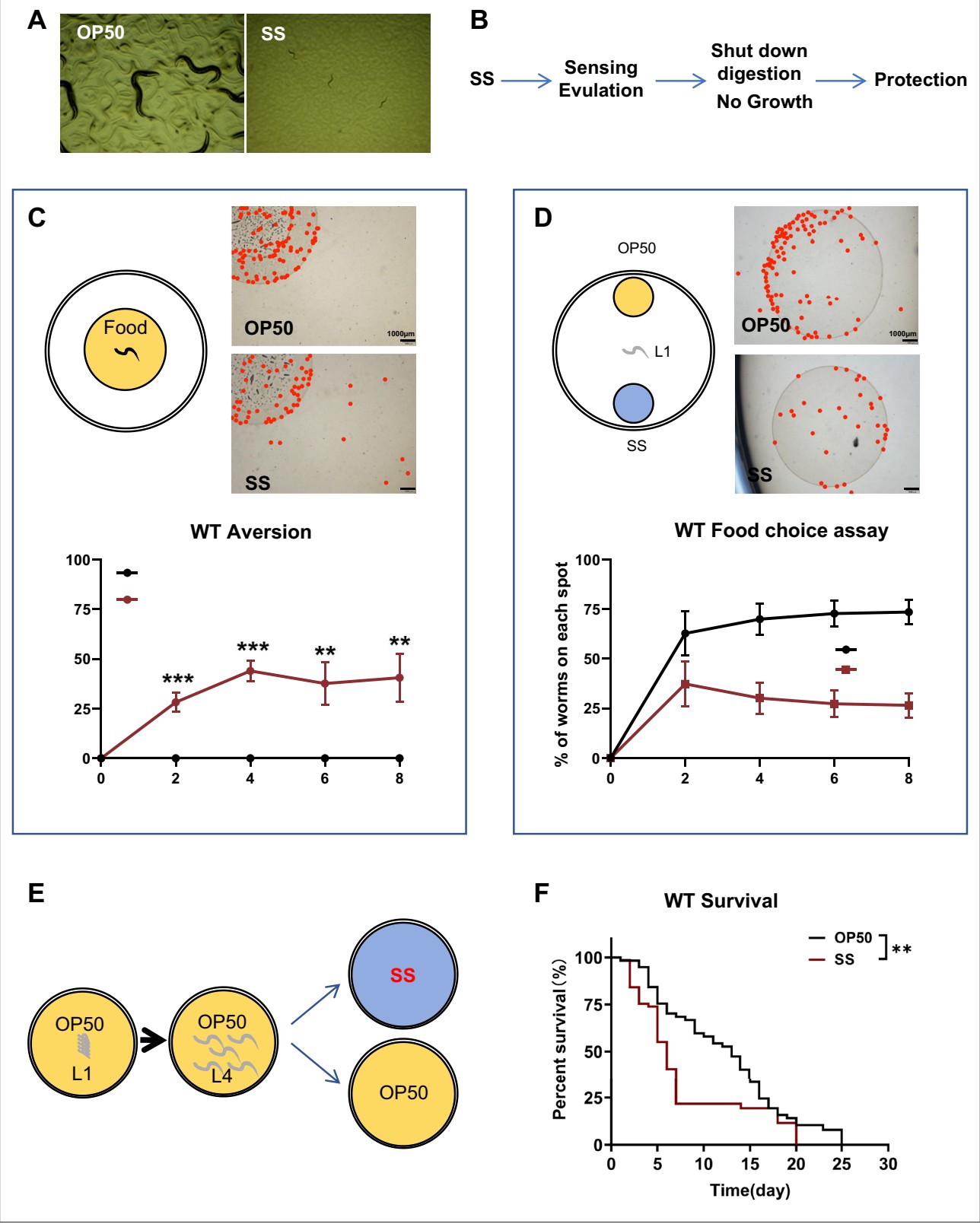

**Figure 1.** *Caenorhabditis elegans* shuts down its digestion for survival when fed harmful food, specifically *Staphylococcus saprophyticus* (SS). (**A**) Microscopic images showing worms fed with SS arrested at the L1 stage 3 days after hatching. (**B**) Schematic model illustrating our hypothesis: *C. elegans* can sense or evaluate inedible food, such as SS, and subsequently shut down their digestion to arrest development as a protective survival strategy. (**C**) Schematic drawing and quantitative data of the food dwelling/avoidance assay. Yellow circles indicate the food spot for OP50 or SS

*Figure 1 continued on next page*

*Figure 1 continued*

bacteria, respectively. The animals were scored at the indicated times after L1 worms were placed on the food spot. The red point indicates the position of each worm. Data are represented as mean ± SD. Scale bar = 1000 µm. \*\*\*p<0.001; \*\*p<0.01 by Student's *t*-test. (**D**) Schematic drawing, microscopic images, and quantitative data of the food choice assay. L1 worms were placed at the center spot (origin). OP50 (yellow) and SS (blue) bacteria were placed on opposite sides of the plate. The red point indicates the position of each worm. The percentage of worms on each spot was calculated at the indicated times. Data are represented as mean ± SD. Scale bar = 1000 µm. \*\*\*\*p<0.0001; \*\*p<0.01 by Student's *t*-test. (**E**, **F**) Schematic drawing and quantitative data of the lifespan of animals fed with SS or OP50. L1 worms were seeded onto OP50 and grown to the L4 stage. L4 worms were then moved to SS or OP50 food to measure lifespan. \*\*p<0.01 by log-rank test. All data are representative of at least three independent experiments.

The online version of this article includes the following source data and figure supplement(s) for figure 1:

**Source data 1.** Numerical data of *Figure 1C–F*.

**Figure supplement 1.** *Staphylococcus saprophyticus* (SS) is harmful food that animals cannot digest.

**Figure supplement 1—source data 1.** Numerical data of *Figure 1—figure supplement 1A–D*.

exhibited a growth phenotype under SS feeding conditions (*Figure 2—figure supplement 1B*). Whole-genome deep sequencing revealed that *ylf6* carries two mutations in the *nsy-1* gene (H929Y, Q1191 [stop codon*]) (*Figure 2—figure supplement 1C*).

To confirm that *nsy-1* is essential for shutting down SS digestion, we used an independent *nsy-1* mutant allele, *ag3*, and found that *nsy-1(ag3)* mutants also digested SS (*Figure 2A*). To determine whether *nsy-1* is crucial for sensing SS, we performed two behavioral assays: food dwelling/avoidance and food choice. In the food dwelling/avoidance assay, larval stage *nsy-1* mutant exhibited significantly impaired avoidance responses at both 4 h and 6 h but not at 8 h, suggesting that NSY-1 is essential for sustained aversion to SS food in the early response (*Figure 2B*). In contrast to wild-type N2 animals, *nsy-1* mutants preferred SS when given a choice between two poor-quality foods, heat-killed *E. coli* and SS (*Figure 2C*, *Figure 1—figure supplement 1B*). These results suggest that *nsy-1* is essential for *C. elegans* to sense and avoid SS.

Next, we examined whether larvae that cannot sense SS and do not shut down digestion can adapt to SS environments. We measured the survival rate of *nsy-1* mutants under SS feeding conditions and found that these mutants had a higher mortality rate (*Figure 2D*).

Overall, these results indicate that *C. elegans* sense and detect unfavorable food, such as SS, through *nsy-1* and subsequently shut down digestion to protect themselves and enhance survival.

## NSY-1 functions in AWC neurons to shut down SS digestion

The *nsy-1* gene in *C. elegans* encodes a MAP kinase kinase kinase (MAPKKK) that operates in the AWC neurons (*Kim et al., 2002*; *Sagasti et al., 2001*), which are essential for chemotaxis and odor sensation. The primary role of *nsy-1* in AWC neurons is to regulate the asymmetric expression of odorant receptors, contributing to neuronal asymmetry and diversity (*Chuang et al., 2007*; *Sagasti et al., 2001*). We hypothesized that the shutdown of SS digestion in *C. elegans* is mediated by *nsy-1* function in AWC neurons.

Firstly, we constructed a P*nsy-1*::GFP reporter strain and confirmed that *nsy-1* is expressed in the AWC neurons (*Figure 3A*). Secondly, we expressed *nsy-1* in the AWC neurons of *nsy-1* mutant animals and observed that the transgenic animals rescued the indigestion phenotype (*Figure 3B*). This indicates that *nsy-1* functions in AWC neurons to shut down SS digestion. Thirdly, we used CRISPR to construct a mutant strain that knocks out *nsy-1* specifically in AWC neurons (*Figure 3—figure supplement 1*). We found that *nsy-1* knockout in AWC neurons also resulted in the shutdown of SS digestion (*Figure 3C*). These results collectively suggest that NSY-1 is functional in AWC neurons and is crucial for shutting down SS digestion in *C. elegans*.

Beyond its established role in AWC neurons, we detected NSY-1 expression in the intestine (*Figure 3—figure supplement 2A*). To assess intestinal NSY-1 function, we performed tissue-specific rescue experiments in *nsy-1* mutants using the intestinal-specific *vha-6* promoter. Intestinal expression of NSY-1 significantly suppressed the enhanced SS digestion phenotype in *nsy-1* mutants (*Figure 3—figure supplement 2B*), demonstrating functional involvement of gut-localized NSY-1 in regulating digestive responses. We propose intestinal NSY-1 mediates this effect through innate immune signaling, consistent with its known pathway components. As previously established (*Geng et al., 2022*), the canonical PMK-1/p38 MAPK pathway functions downstream of NSY-1, with both *sek-1* and *pmk-1* knockdown enhancing SS digestion through immune modulation. This indicates intestinal

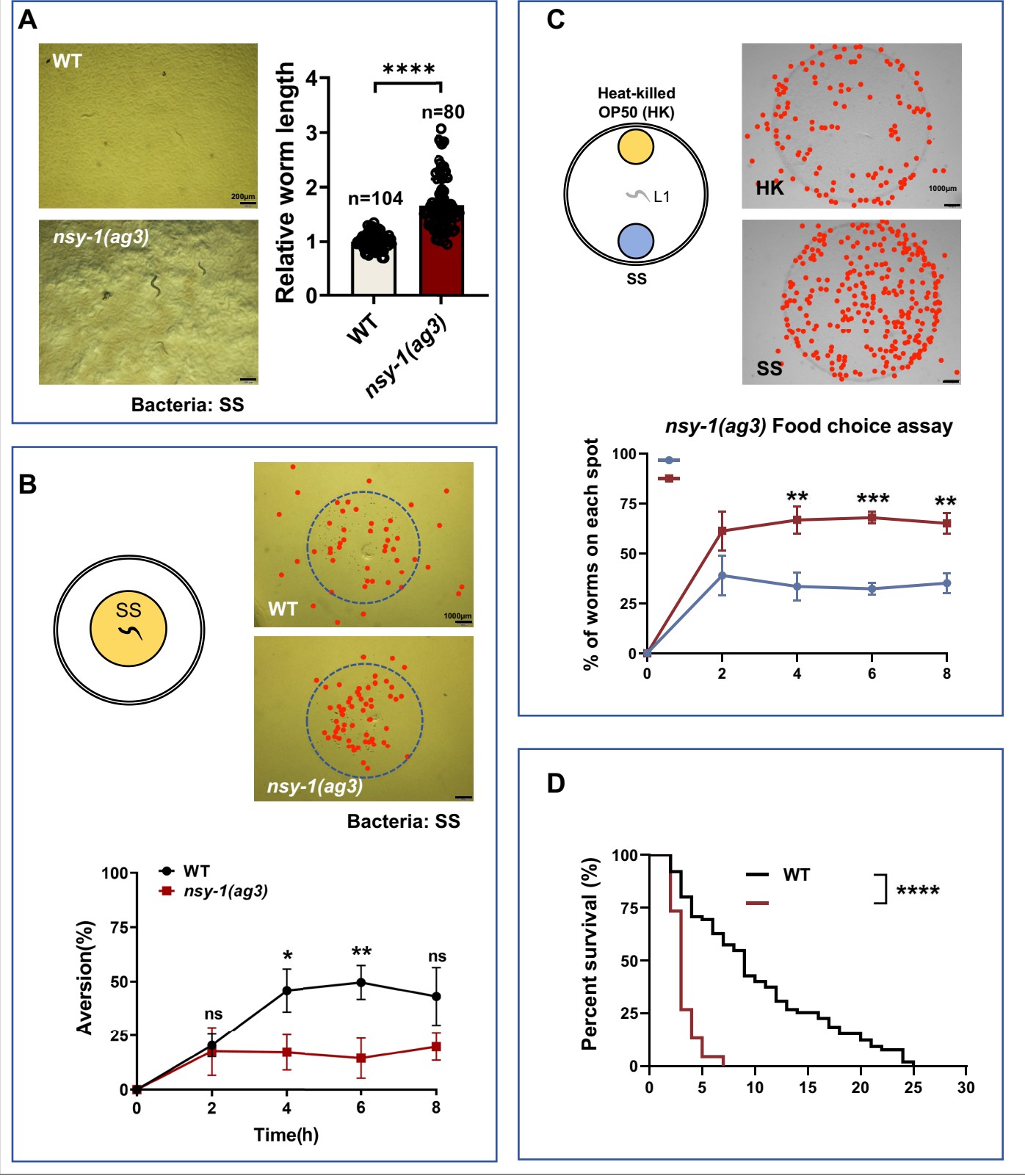

**Figure 2.** *Caenorhabditis elegans* senses *Staphylococcus saprophyticus* (SS) and shuts down digestion through NSY-1. (**A**) Developmental phenotype of wild-type N2 and *nsy-1(ag3)* mutant worms fed with SS bacteria. Data are represented as mean ± SD. Scale bar = 200 µm. ****p<0.0001 by Student's *t*-test. n = number of animals which were scored. (**B**) Schematic drawing, microscopic images, and quantitative data of the food dwelling/avoidance assay. Yellow circles indicate the food spot for SS bacteria. The animals were scored at the indicated times after L1 worms were placed on the food spot.

*Figure 2 continued on next page*

*Figure 2 continued*

The blue circle indicates the edge of the bacterial lawn, and the red point indicates the position of each worm. Data are represented as mean ± SD. Scale bar = 1000 μm. *p<0.05; **p<0.01 by Student's *t*-test. (**C**) Schematic drawing, microscopic images, and quantitative data of the food choice assay. L1 *nsy-1(ag3)* worms were placed at the center spot (origin). Heat-killed OP50 (yellow) and SS (blue) bacteria were placed on opposite sides of the plate. The red point indicates the position of each worm. The percentage of worms on each spot was calculated at the indicated times. Data are represented as mean ± SD. Scale bar = 1000 μm. **p<0.01; ***p<0.001 by Student's *t*-test. (**D**) Survival curves of wild-type N2 and *nsy-1(ag3)* mutant worms fed with SS bacteria. L4 worms, previously fed OP50 bacteria, were transferred to SS food to measure lifespan. ****p<0.0001 by log-rank test. All data are representative of at least three independent experiments.

The online version of this article includes the following source data and figure supplement(s) for figure 2:

**Source data 1.** Numerical data of *Figure 2A–D*.

**Figure supplement 1.** Ethyl methanesulfonate (EMS) screen to identify genes involved in shutting down digestion of *Staphylococcus saprophyticus* (SS).

NSY-1 suppresses digestion may act through PMK-1-mediated immune responses. Since neuronal NSY-1's role in digestive control was previously undefined, we prioritized mechanistic analysis of its neuronal function in digestion regulation.

To determine whether NSY-1 in AWC neurons mediates SS sensory perception, we performed dwelling (avoidance) and food-choice assays using AWC-specific *nsy-1* knockout and AWC-rescued strains (*nsy-1(ag3)*; P*odr-1::nsy-1*). In dwelling assays, AWC-specific *nsy-1* KO mutants exhibited significantly impaired SS avoidance at 6 h (**Figure 3—figure supplement 3A**), while AWC-rescued strains restored avoidance capacity at 2–6 h (**Figure 3—figure supplement 3B**). Food-choice assays further revealed that AWC *nsy-1* KO mutants preferentially migrated toward SS (**Figure 3—figure supplement 3C**), whereas AWC-rescued showed no preference between SS and HK-*E. coli* (**Figure 3—figure supplement 3D**). These data conclusively demonstrate that NSY-1 acts in AWC neurons to mediate SS recognition and aversion behaviors.

## AWC neurons exhibit OFF state in sensing SS food

In *C. elegans*, the expression of the *str-2* gene in AWC neurons indicates the ON state (AWC^ON) (**Sagasti et al., 2001**), which is associated with high cGMP levels and lower calcium activity, enabling the neuron to respond to specific odors. Conversely, the absence of *str-2* expression marks the OFF state (AWC^OFF), characterized by different odor responses, low cGMP levels, and higher calcium activity (**Troemel et al., 1999**). We aimed to investigate (1) whether AWC neurons exhibit different states under normal food (*E. coli* OP50) versus unfavorable food (SS) conditions and (2) whether the state of AWC neurons affects the ability of *C. elegans* to digest SS.

Using *str-2*::GFP as a marker for AWC neuron states (**Troemel et al., 1999**), we found that wild-type animals feeding on normal OP50 food typically exhibit one AWC^OFF and one AWC^ON neuron. However, when feeding on SS, the proportion of animals with the AWC^OFF state increased, with approximately 50% of animals exhibiting two AWC^OFF neurons (**Figure 3D–F**).

In *nsy-1* mutant animals feeding on OP50, *str-2*::GFP is expressed in both AWC neurons (2AWC^ON), consistent with previous studies (**Chuang and Bargmann, 2005**; **Chuang et al., 2007**; **Sagasti et al., 2001**; **Troemel et al., 1999**; **Figure 3D and E**). Notably, both AWC neurons remained in the AWC^ON state in *nsy-1* mutants feeding on SS (**Figure 3D and E**). These results suggest that SS feeding induces an AWC^OFF state in wild-type animals, while the *nsy-1* mutation promotes an AWC^ON state even under SS feeding conditions. This implies that the AWC^OFF state may inhibit SS digestion, whereas the AWC^ON state promotes it.

The TIR-1-NSY-1-SEK-1-MAPK pathway plays a crucial role in regulating the asymmetric AWC cell fate decision. Previous studies have shown that *str-2*::GFP expression in both AWC cells (2AWC^ON phenotype) occurs in *tir-1*, *nsy-1*, and *sek-1* mutant animals (**Chuang and Bargmann, 2005**; **Sagasti et al., 2001**; **Troemel et al., 1999**). We found that *tir-1* mutant can grow under SS feeding conditions (**Figure 3G**), indicating that *tir-1* mutant can digest SS similarly to *nsy-1* mutants. This demonstrates that the AWC^ON state promotes SS digestion.

To confirm the importance of AWC state in SS digestion, we performed AWC-specific neuron ablation experiments using previously validated transgenic strain that expresses cleaved caspase under the AWC-specific promoter, *ceh-36* (*ceh-36p*::caspase). Critically, worms with ablated AWC neurons completely failed to digest SS food (**Figure 3—figure supplement 4**), phenocopying the non-digesting

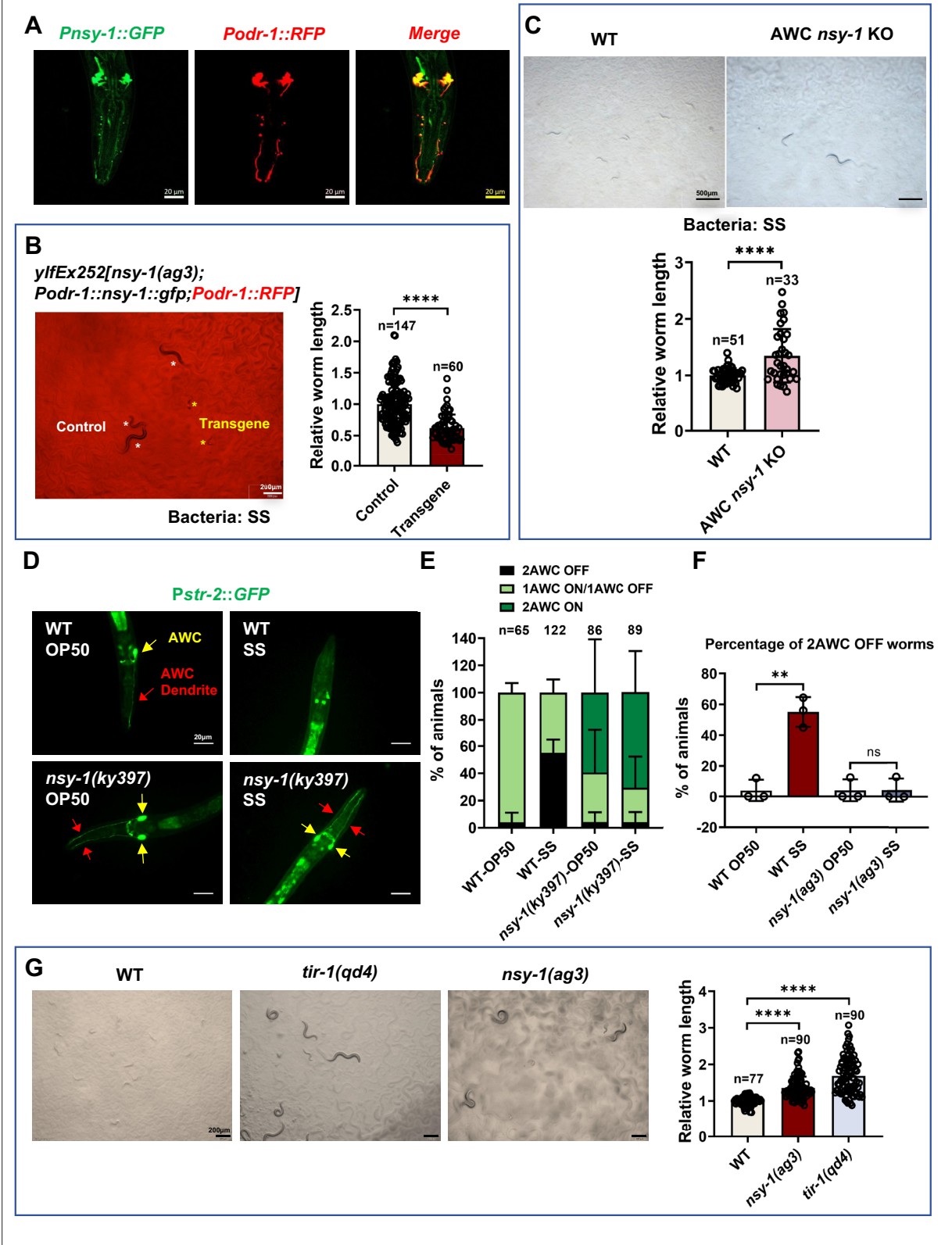

**Figure 3.** NSY-1 plays a critical role in AWC neurons to inhibit *Staphylococcus saprophyticus* (SS) digestion. (**A**) Microscopic image showing the expression pattern of *nsy-1*. The head of an adult transgenic animal carrying *Pnsy-1::GFP* and *Podr-1::RFP* shows colocalization of *nsy-1* and *odr-1*. Scale bar = 20 μm. (**B**) Developmental progression of *nsy-1(ag3)* mutant worms carrying *Podr-1::nsy-1::gfp* (AWC neuron-specific expression) grown on SS bacteria. Control animals are labeled with white stars, and animals carrying the transgenes (rescued animals) are labeled with yellow stars. Data

*Figure 3 continued on next page*

*Figure 3 continued*

are represented as mean ± SD. Scale bar = 200 µm. ****p<0.0001 by Student's *t*-test. n=number of animals which were scored. (**C**) Developmental progression of wild-type N2 and AWC neuron-specific knockout *nsy-1* animals (AWC *nsy-1* KO) grown on SS bacteria. Data are represented as mean ± SD. Scale bar = 500 µm. ***p<0.001 by Student's *t*-test. n=number of animals which were scored. (**D**) Microscopic images show P*str-2::GFP*, a marker for AWC neuron states, in L1-staged wild-type and *nsy-1(ky397)* mutant worms grown on OP50 or SS bacteria for 6 h. AWC neuron positions are highlighted with red and yellow arrows. Scale bar = 20 µm. (**E, F**) Percentage of animals with different AWC neuron states. *nsy-1* mutation promotes a 2AWC$^{ON}$ state under SS feeding conditions (**E**), with approximately 50% of animals exhibiting 2AWC$^{OFF}$ neurons when feeding on SS (**F**). Data are represented as mean ± SD. **p<0.01 by Student's *t*-test. n=number of animals which were scored. (**G**) Developmental progression of wild-type N2, *tir-1(qd4)*, and *nsy-1(ag3)* mutant worms grown on SS bacteria. Data are represented as mean ± SD. Scale bar = 200 µm. ****p<0.0001 by Student's *t*-test. n=number of animals which were scored. All data are representative of at least three independent experiments.

The online version of this article includes the following source data and figure supplement(s) for figure 3:

**Source data 1.** Numerical data of *Figure 3B–G*.

**Figure supplement 1.** Construction of *nsy-1*-specific knockout in AWC neurons using CRISPR-Cas9.

**Figure supplement 1—source data 1.** Original gels for *Figure 3—figure supplement 1*, indicating the relevant bands and treatments.

**Figure supplement 1—source data 2.** Original files for gels analysis displayed in *Figure 3—figure supplement 1*.

**Figure supplement 2.** NSY-1 functions in the intestine to shut down *Staphylococcus saprophyticus* (SS) digestion.

**Figure supplement 2—source data 1.** Numerical data of *Figure 3—figure supplement 2B*.

**Figure supplement 3.** NSY-1 functions in AWC neurons to influence the recognition of *Staphylococcus saprophyticus* (SS).

**Figure supplement 3—source data 1.** Numerical data of *Figure 3—figure supplement 3A–D*.

**Figure supplement 4.** AWC neurons are essential for initiating *Staphylococcus saprophyticus* (SS) digestion.

**Figure supplement 4—source data 1.** Numerical data of *Figure 3—figure supplement 4*.

state of wild-type worms on SS. This result directly confirms that functional AWC neurons are essential for initiating SS digestion, aligning with our model where the AWC-OFF state (induced by SS) inhibits digestion while the AWC-ON state promotes it.

Overall, our data suggest that the state of AWC neurons is critical for sensing food in *C. elegans*. When sensing SS food, animals exhibit an AWC$^{OFF}$ state, which shuts down digestion. Conversely, the AWC$^{ON}$ state promotes the digestion of SS.

## NSY-1 shuts down SS digestion through induction of STR-130

We have demonstrated that NSY-1 in AWC neurons detects unfavorable food and shuts down digestion (*Figure 3*). To investigate the genes regulated by NSY-1 in response to SS and their impact on SS digestion, we conducted a transcriptomic analysis on L1 larval animals fed with normal food (OP50) or unfavorable food (SS) for a short duration (4 h). We speculated that some genes induced by SS food are dependent on NSY-1 (*Figure 4A*), and their induction aids in shutting down SS digestion.

RNA-seq data analysis revealed 304 NSY-1-dependent candidate genes responding to SS (*Figure 4B*, *Supplementary file 1*). Enrichment analysis (*Figure 4—figure supplement 1A*, *Supplementary file 2*) of these candidate genes showed mainly associations with biotic stimulus, defense responses, xenobiotic stimulus, suggesting that NSY-1 positively regulates stress response pathways to protect animals under harmful food, SS, feeding conditions. Moreover, we found that sensory perception-related genes (*sra-32*, *str-87*, *str-112*, *str-130*, *str-160*, *str-230*) (*Figure 4—figure supplement 1A*, *Supplementary file 2*) were also enriched, with many genes being G protein-coupled receptors (GPCRs), which mediate odor sensing (*Buck and Axel, 1991*).

We further analyzed the dependence of these enriched GPCRs on NSY-1 under SS feeding conditions (*Figure 4—figure supplement 1B*) and found that *str-130* is significantly upregulated in response to SS, with its function strongly being NSY-1 dependent (*Figure 4C*).

Using RNAi knockdown and the SS growth assay, we observed that RNAi of *str-130*, *str-230*, *str-87*, or *str-112* significantly enhanced SS growth (*Figure 4—figure supplement 2A*), with *str-130* RNAi exhibiting the most robust phenotype—phenocopying *nsy-1(ag3)* mutants. Crucially, none of these GPCR knockdowns further enhanced growth in *nsy-1(ag3)* mutants (*Figure 4—figure supplement 2B*), confirming their position downstream of NSY-1. These data establish *str-130* as the dominant effector of NSY-1-mediated SS response regulation, while suggesting minor contributions from other GPCRs (*str-230*, *str-87*, *str-112*).

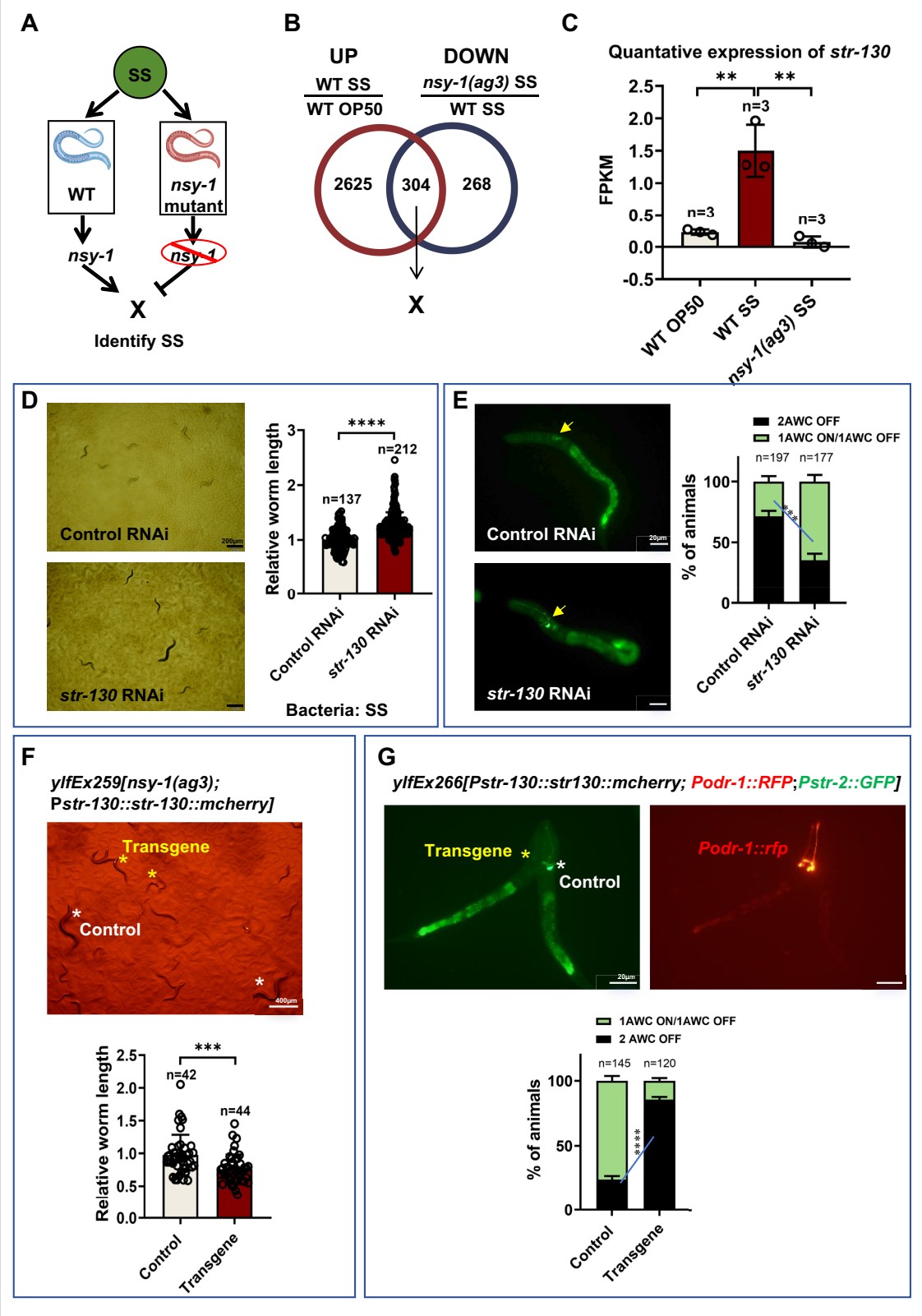

**Figure 4.** NSY-1 inhibits animals from digesting *Staphylococcus saprophyticus* (SS) by inducing *str-130*. (**A**) Schematic illustration showing that 'X' genes rely on NSY-1 to shut down SS digestion. 'X' genes induced by SS food are dependent on NSY-1, and their induction aids in shutting down SS digestion. (**B**) Venn diagram showing the overlap of genes that respond to SS and rely on NSY-1. The number of genes is indicated in the diagram (also see ***Supplementary file 1***). (**C**) Transcriptome analysis showing *str-130* mRNA expression, which relies on NSY-1 in response to SS. Data are represented

*Figure 4 continued on next page*

*Figure 4 continued*

as mean ± SD. **p<0.01 by Student's *t*-test. (**D**) Developmental progression of wild-type animals treated with control RNAi or *str-130* RNAi grown on SS bacteria. Data are represented as mean ± SD. Scale bar = 200 μm. ****p<0.0001 by Student's *t*-test. n=number of animals which were scored. (**E**) Microscopic images and quantitative data of AWC neuron states in L1 animals treated with control RNAi or *str-130* RNAi grown on SS bacteria. Data are represented as mean ± SD. Scale bar = 20 μm. ***p<0.001 by Student's *t*-test (1AWC$^{ON}$/1AWC$^{OFF}$: control vs *str-130* RNAi). n=number of animals which were scored. (**F**) Developmental progression of *nsy-1(ag3)* mutant worms carrying *Pstr-130::str-130::mCherry* grown on SS bacteria. Control animals are labeled with white stars, and animals carrying transgenes are labeled with yellow stars. Data are represented as mean ± SD. Scale bar = 400 μm. ***p<0.001 by Student's *t*-test. n=number of animals which were scored. (**G**) Microscopic images and quantitative data of AWC neuron states in L1 animals carrying *Pstr-130::str-130::mCherry*. Transgenic animals with overexpression of *str-130* (carrying *Pord-1::GFP* as a co-injection marker) show an increased 2AWC$^{OFF}$ state. Data are represented as mean ± SD. Scale bar = 20 μm. ****p<0.001 by Student's *t*-test (2AWC$^{OFF}$: Control vs Transgene). n=number of animals which were scored. All data are representative of at least three independent experiments.

The online version of this article includes the following source data and figure supplement(s) for figure 4:

**Source data 1.** Numerical data of *Figure 4B–G*.

**Figure supplement 1.** *str-130* is induced in wild-type in response to *Staphylococcus saprophyticus* (SS), dependent on NSY-1.

**Figure supplement 1—source data 1.** Numerical data of *Figure 4—figure supplement 1A and B*.

**Figure supplement 2.** *str-130* as the dominant effector of NSY-1-mediated *Staphylococcus saprophyticus* (SS) response regulation.

**Figure supplement 2—source data 1.** Numerical data of *Figure 4—figure supplement 2A, B*.

**Figure supplement 3.** NSY-1 inhibits *Staphylococcus saprophyticus* (SS) digestion by inducing GPCR *str-130*.

**Figure supplement 3—source data 1.** Numerical data of *Figure 4—figure supplement 3*.

It has been shown that *str-130* is expressed in AWC$^{OFF}$ neurons, based on transgenic GFP reporter strains, *str-130p*::GFP (*Vidal et al., 2018*). Our data also show that *str-130* expression is induced in wild-type animals fed with SS (*Figure 4C*), where AWC neurons exhibit the AWC$^{OFF}$ state (*Figure 3D and E*). Therefore, it is possible that the high expression of *str-130*, regulated by *nsy-1*, alters the AWC state and inhibits SS digestion.

Firstly, we found that knockdown of *str-130* in wild-type animals promoted SS digestion, thereby supporting animal growth (*Figure 4D*), and the proportion of animals with two AWC$^{OFF}$ neurons decreased (*Figure 4E*). Secondly, we found that overexpression of *str-130* in *nsy-1* mutant animals inhibited SS digestion, thereby slowing animal growth (*Figure 4F*), and the proportion of animals with two AWC$^{OFF}$ neurons increased (*Figure 4G*). These results demonstrate that NSY-1 promotes the AWC$^{OFF}$ state by inducing *str-130* expression, which in turn inhibits SS digestion in *C. elegans*.

To definitively establish the epistatic relationship between NSY-1 and STR-130, we performed RNAi knockdown of *str-130* in the *nsy-1(ag3)* mutant background and assessed development on SS food. We found that the *str-130* RNAi did not further enhance the developmental capacity of *nsy-1(ag3)* mutant animals on SS (*Figure 4—figure supplement 3*). This epistasis confirms STR-130 functions strictly downstream of NSY-1 within the same genetic pathway. Together with our overexpression data (*Figure 4F and G*) showing neuronal *str-130* rescue suppresses SS digestion in *nsy-1* mutants, these results establish a linear signaling axis where NSY-1 primarily achieves functional inhibition of SS digestion through induction of the GPCR *str-130*.

## NSY-1 mutation promotes SS digestion by inducing insulin signaling

NSY-1 mutation promotes the digestion of unfavorable food, such as SS, and supports *C. elegans* growth (*Figure 2A*). We hypothesize that, in addition to upregulating certain genes, such as *str-130*, to inhibit SS digestion, NSY-1 may also suppress certain genes to prevent nematodes from utilizing SS (*Figure 5A*). One possibility is that some genes, induced by the *nsy-1* mutation under SS feeding conditions, could facilitate SS digestion in the *nsy-1* mutant (*Figure 5A*).

Our RNA-seq analysis identified 308+46 = 354 genes that are induced by the *nsy-1* mutation under SS feeding conditions (*Figure 5B*, *Supplementary file 3*). However, among these 354 genes, 46 genes can also be induced in wild-type animals fed with SS, suggesting that these 46 genes may not be involved in digesting SS in *nsy-1* mutants. Therefore, the 308 candidate genes induced by the *nsy-1* mutation under SS feeding conditions could potentially promote animals to digest SS.

Enrichment analysis revealed that genes related to extracellular functions, such as insulin-related genes, are induced in *nsy-1* mutant animals (*Figure 5—figure supplement 1A*, *Supplementary file 4*). Further analysis of insulin-related genes from the RNA-seq data showed that *ins-23* is predominantly

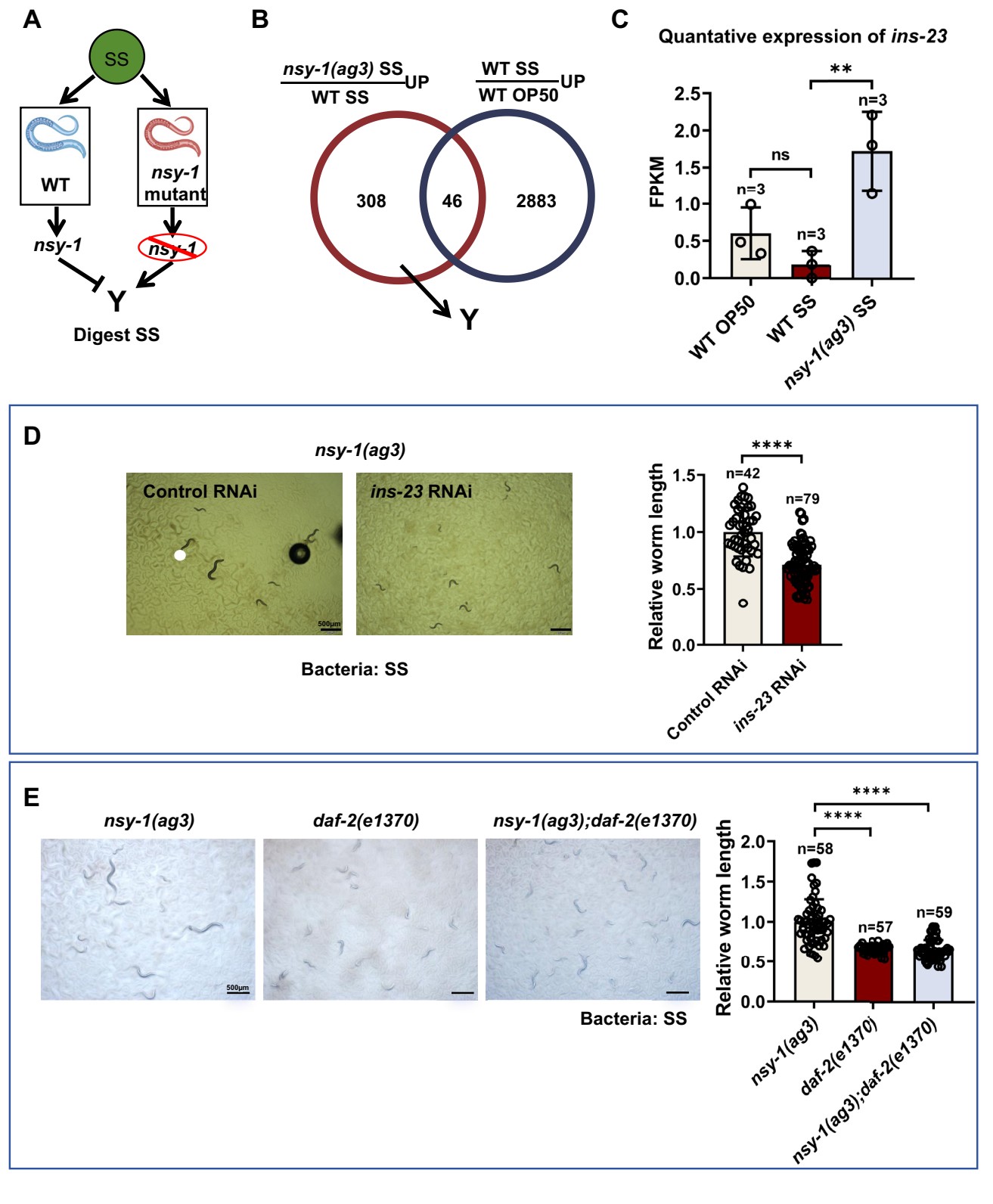

**Figure 5.** NSY-1 mutation activates animals to digest *Staphylococcus saprophyticus* (SS) by inducing insulin signaling. (**A**) Schematic illustration showing that NSY-1 inhibits the expression of 'Y' genes, which promote SS digestion. Some genes induced by the *nsy-1* mutation under SS feeding conditions could facilitate SS digestion in the *nsy-1* mutant. (**B**) Venn diagram showing the overlap of genes that respond to SS but are limited by NSY-1. A total of 308 candidate genes induced by the *nsy-1* mutation under SS feeding conditions could potentially promote SS digestion. (**C**) Transcriptome analysis

*Figure 5 continued on next page*

*Figure 5 continued*

showing that *ins-23* expression is induced in animals with the *nsy-1* mutation under SS feeding conditions. Data are represented as mean ± SD. **p<0.01; n.s., not significant by Student's t-test. n=3 biological replicates. (**D**) Developmental progression of *nsy-1(ag3)* mutant animals treated with control RNAi or *ins-23* RNAi grown on SS bacteria. Data are represented as mean ± SD. Scale bar = 500 μm. ****p<0.0001 by Student's *t*-test. n=number of animals which were scored. (**E**) Developmental progression of *nsy-1(ag3), daf-2(e1370)*, and *nsy-1(ag3);daf-2(e1370)* double mutant animals grown on SS bacteria. Data are represented as mean ± SD. Scale bar = 500 μm. ****p<0.0001 by Student's *t*-test. n=number of animals which were scored. All data are representative of at least three independent experiments.

The online version of this article includes the following source data and figure supplement(s) for figure 5:

**Source data 1.** Numerical data of *Figure 5B–E*.

**Figure supplement 1.** *nsy-1* mutation induces the expression of insulin-related genes.

**Figure supplement 1—source data 1.** Numerical data of *Figure 5—figure supplement 1A and B*.

**Figure supplement 2.** NSY-1 mutation promotes *Staphylococcus saprophyticus* (SS) digestion by inducing *ins-23*.

**Figure supplement 2—source data 1.** Numerical data of *Figure 5—figure supplement 2*.

**Figure supplement 3.** INS-23 induction in *nsy-1* mutants promotes digestion independently of intestinal DAF-2 function.

**Figure supplement 3—source data 1.** Numerical data of *Figure 5—figure supplement 3B and C*.

induced in *nsy-1* mutant animals (*Figure 5—figure supplement 1B*), suggesting its potential role in promoting SS digestion.

To determine if insulin-like peptide genes were functionally responsible for the enhanced SS growth observed in *nsy-1(ag3)* mutants, we performed functional phenotypic screening using the SS growth assay (worm length assay). We individually knocked down each of these candidates (*ins-22, ins-23, ins-24, ins-27*) in the *nsy-1(ag3)* mutant background. Among these, only RNAi targeting *ins-23* significantly suppressed the enhanced development of the *nsy-1(ag3)* mutant on SS (*Figure 5—figure supplement 2*, *Figure 5D*). This targeted functional screening revealed that *ins-23* has the most robust and specific role in mediating the enhanced digestion phenotype downstream of NSY-1 loss, providing the critical justification for our subsequent focus on this particular insulin-like peptide.

Given that INS-23 is expressed in AWC neurons (*Figure 5—figure supplement 3A*, from CeNGEN), this suggests increased production and likely enhanced release of INS-23 from AWC neurons in the *nsy-1(ag3)* mutant background, which promotes SS digestion.

The insulin/insulin-like growth factor signaling (IIS) pathway, particularly through the DAF-2 receptor, integrates nutritional signals to regulate various behavioral and physiological responses related to food (*Kodama et al., 2006*; *Ryu et al., 2018*). It has been shown that INS-23 acts as an antagonist for the DAF-2 receptor to promote larval diapause (*Matsunaga et al., 2018*). To test whether *ins-23* induction in *nsy-1(ag3)* mutants promotes SS digestion through its receptor, DAF-2, we constructed a *nsy-1; daf-2* double mutant. We found that the SS digestion ability of the *nsy-1* mutant was inhibited by the *daf-2* mutation (*Figure 5E*). This suggests that the *nsy-1* mutation induces the insulin peptide *ins-23*, which promotes SS digestion through its potential receptor, DAF-2.

To investigate whether DAF-2 acts as the gut-localized receptor for neuronal INS-23 signaling, we performed tissue-specific rescue experiments in the *nsy-1(ag3);daf-2(e1370)* double mutant. When DAF-2 was re-introduced specifically in the intestine (using the *ges-1* promoter), we observed a significant suppression of SS digestion (*Figure 5—figure supplement 3B*), but not rescue digestive defect. This indicates that INS-23 induction in *nsy-1* mutants promotes digestion independently of intestinal DAF-2 function.

As established in our prior work (*Geng et al., 2022*), SS exposure triggers phosphorylation of PMK-1 (P-PMK-1) in *C. elegans*, and *pmk-1* mutants exhibit enhanced growth on SS. This confirms that PMK-1-mediated innate immune signaling actively regulates SS responsiveness and digestion. To address whether PMK-1 functions downstream of NSY-1 within our proposed model, we performed critical epistasis analyses. While we observed that *nsy-1* mutation elevates *ins-23* (indicating NSY-1 suppression of *ins-23*), knockdown of *pmk-1* did not alter *ins-23* expression levels (*Figure 5—figure supplement 3C*). This demonstrates that PMK-1 does not operate through the INS-23 pathway to regulate SS digestion. Thus, although both pathways respond to SS, the PMK-1-mediated innate immune response and the NSY-1/INS-23 axis constitute distinct regulatory mechanisms governing digestive adaptation.

## NSY-1 mutation promotes SS digestion through regulation of intestinal BCF-1

In our previous study, we found that heat-killed *E. coli* promotes SS digestion in *C. elegans* (**Geng et al., 2022**), a process requiring intestinal BCF-1 (**Hao et al., 2024**). In the absence of BCF-1, the digestive capability of the animals is significantly reduced (**Hao et al., 2024**). This led us to investigate whether NSY-1 in AWC neurons regulates intestinal *bcf-1* expression.

Firstly, we used a BCF-1::GFP reporter to measure *bcf-1* expression in *nsy-1* mutant animals. We found that mutation of *nsy-1* induced *bcf-1* expression in animals fed either SS or OP50 food (**Figure 6A**). Additionally, we confirmed that *nsy-1* mutation in AWC neurons also induced intestinal *bcf-1* expression under SS feeding conditions (**Figure 6B**). This data indicates that NSY-1 in AWC neurons inhibits intestinal *bcf-1* expression, implying that *nsy-1* mutation promotes SS digestion through BCF-1.

Next, we constructed a *nsy-1; bcf-1* double mutant and analyzed the growth of these animals on SS. We found that the digestive ability of *nsy-1* mutants was inhibited by the mutation of *bcf-1* (**Figure 6C**). Together, our data suggest that the increased digestion ability in *nsy-1* mutant animals is dependent on the intestinal digestion factor BCF-1.

We then asked how *nsy-1* regulates intestinal *bcf-1* expression. Since *nsy-1* mutation induces the insulin peptide *ins-23*, which promotes SS digestion, we tested whether the induction of intestinal *bcf-1* by *nsy-1* mutation is also mediated through INS-23. We found that the *bcf-1*::GFP level decreased in *nsy-1* mutant animals following *ins-23* RNAi treatment (**Figure 6D**). This suggests that *nsy-1* mutation activates *bcf-1* expression in the intestine, which requires INS-23.

To test whether INS-23 acts in AWC neurons to regulate intestinal BCF-1, we generated AWC-specific knockdown strains, which were achieved by rescuing *sid-1* cDNA expression under the *ceh-36* promoter in a *sid-1(qt9)*;BCF-1::GFP background.

We found that AWC-restricted *ins-23* knockdown significantly reduced intestinal BCF-1::GFP expression (**Figure 6—figure supplement 1A**). This confirms that INS-23 functions within AWC sensory neurons to activate intestinal BCF-1, consistent with NSY-1's upstream inhibition of INS-23 in this neuronal subtype.

NSY-1 promotes the AWC^OFF^ state through STR-130 to suppress SS digestion. To determine whether AWC-expressed STR-130 regulates intestinal BCF-1 expression, we observed that AWC neuron-specific RNAi of *str-130* elevated intestinal BCF-1::GFP expression (**Figure 6—figure supplement 1B**). This demonstrates that STR-130 functions in AWC neurons to repress BCF-1 expression in the intestine.

## Discussion

This study in *C. elegans* reveals a neural-digestive mechanism for evaluating harmful food (**Figure 7**). The neuron-expressed NSY-1 protein detects SS as unsafe food, triggering a digestive shutdown via the AWC^OFF^ neural circuit and the NSY-1-dependent STR-130. Mutations in NSY-1 lead to SS digestion, activating the insulin/IGF-1 signaling (IIS) pathway and BCF-1 expression. These findings highlight a food quality evaluation strategy where neurons communicate with the digestive system to assess food safety, providing insights into how animals adapt to toxic food environments.

The identification of NSY-1 in AWC neurons as a key player in detecting SS and initiating digestive shutdown is particularly intriguing. NSY-1, a MAP kinase (MAPKKK), is known to regulate the asymmetric expression of odorant receptors, contributing to neuronal asymmetry and diversity (**Chuang and Bargmann, 2005**; **Chuang et al., 2007**; **Sagasti et al., 2001**; **Troemel et al., 1999**). Previous research has demonstrated that AWC neurons are pivotal in chemotaxis and sensory processing (**Bargmann et al., 1993**; **Troemel et al., 1997**; **Wes and Bargmann, 2001**). Our findings extend its function to include food quality assessment, showing that NSY-1 can trigger a digestive shutdown via the AWC^OFF^ neural circuit.

This neural circuit appears to be a crucial component of a systemic food quality control mechanism, allowing animals to adapt effectively to harmful food sources. The state of AWC neurons (AWC^ON^ or AWC^OFF^) directly influences the animal's ability to digest SS, with the AWC^OFF^ state inhibiting digestion and the AWC^ON^ state promoting it. The finding that activation of the AWC^OFF^ neural circuit leads to a systemic digestive shutdown mediated by NSY-1-dependent GPCR(STR-130) is another notable

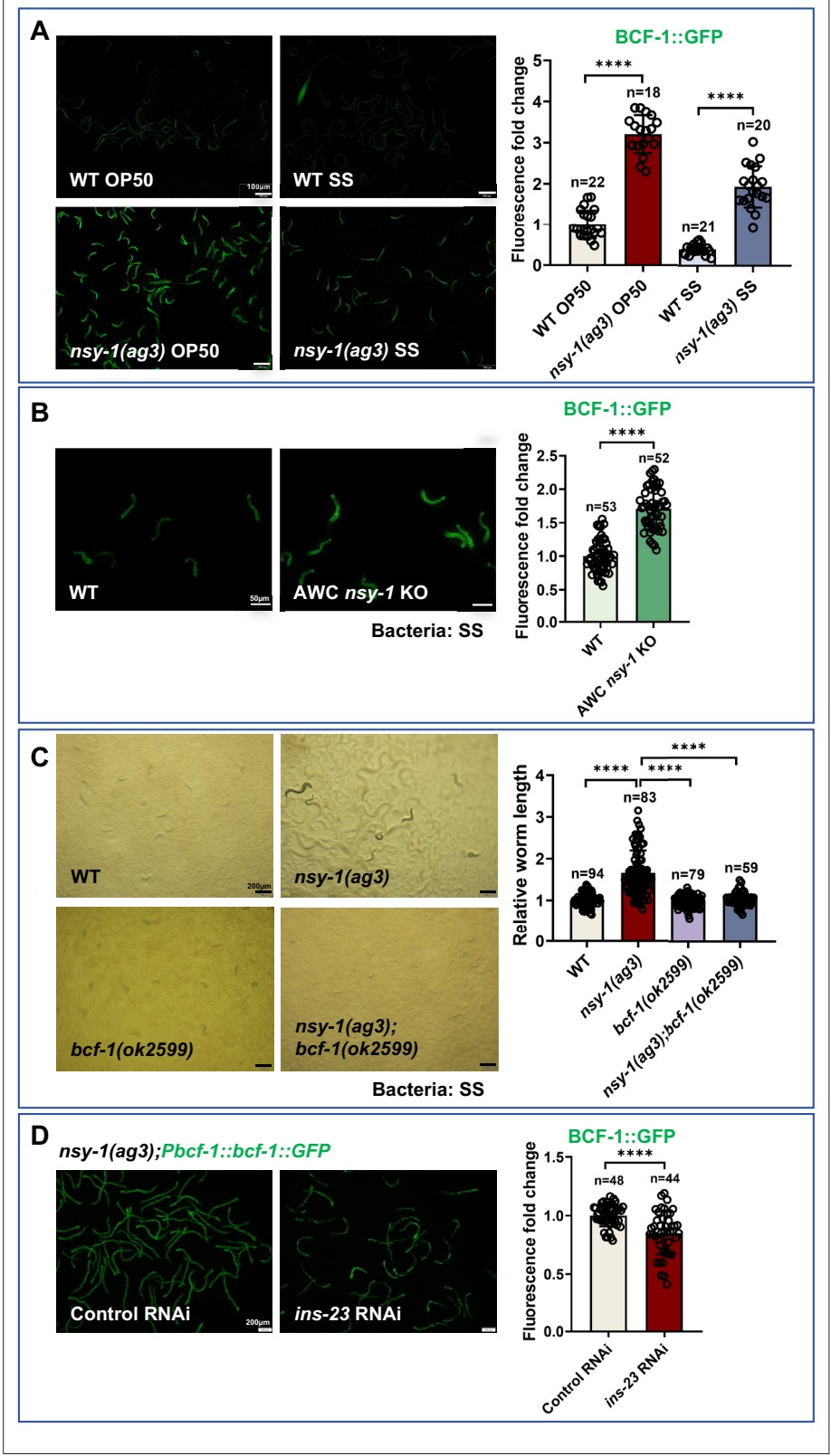

**Figure 6.** NSY-1 mutation promotes animals to digest *Staphylococcus saprophyticus* (SS) through inducing intestinal *bcf-1*. (**A**) Microscopic images and quantitative data showing fluorescence of *Pbcf-1::bcf-1::GFP* in L1-staged wild-type (WT) and *nsy-1(ag3)* mutant animals fed with OP50 or SS bacteria for 6 h. Data are represented as mean ± SD. Scale bar = 100 μm. ****p<0.0001 by Student's *t*-test. n=number of animals which were scored.

*Figure 6 continued on next page*

*Figure 6 continued*

(**B**) Microscopic images and quantitative data showing fluorescence of *Pbcf-1::bcf-1::GFP* in L1-staged wild-type and AWC *nsy-1* KO mutant (AWC neuron-specific knockout *nsy-1* animals) fed with SS bacteria for 6 h. Data are represented as mean ± SD. Scale bar = 50 μm. ****p<0.0001 by Student's *t*-test. n=number of animals which were scored. (**C**) Developmental progression of wild-type N2, *nsy-1(ag3)*, *bcf-1(ok2599)*, and *nsy-1(ag3);bcf-1(ok2599)* double mutant animals grown on SS bacteria. Data are represented as mean ± SD. Scale bar = 200 μm. ****p<0.0001 by Student's *t*-test. n=number of animals which were scored. (**D**) Microscopic images and quantitative data showing fluorescence of *Pbcf-1::bcf-1::GFP* in *nsy-1(ag3)* mutant animals treated with control RNAi or *ins-23* RNAi under normal RNAi feeding conditions. Data are represented as mean ± SD. Scale bar = 200 μm. ****p<0.0001 by Student's *t*-test. n=number of animals which were scored. All data are representative of at least three independent experiments.

The online version of this article includes the following source data and figure supplement(s) for figure 6:

**Source data 1.** Numerical data of *Figure 6A–D*.

**Figure supplement 1.** INS-23 and STR-130 are functions in AWC neurons to affect BCF-1 expression.

**Figure supplement 1—source data 1.** Numerical data of *Figure 6—figure supplement 1A and B*.

advancement. GPCRs are well-known for their roles in sensory perception (*Julius and Nathans, 2012*; *Troemel et al., 1995*). Our results suggested that GPCR STR-130 plays a role in shutting down digestive processes while maintaining AWC^OFF states for evaluating harmful food.

Our study underscores the critical role of insulin signaling pathways in mediating the effects of neuronal detection on intestinal functions. Upon detection of SS by AWC neurons, NSY-1 inhibits the expression of insulin-like peptides, particularly INS-23. These neuronal peptides promote the expression of BCF-1, a key regulatory factor in digestion (*Hao et al., 2024*). Once *ins-23* was inhibited by neuronal NSY-1, the intestinal BCF-1 level is also reduced, which in turn shut down digestion. We found that the digestive ability of *nsy-1* mutants was totally inhibited by the mutation of *bcf-1* (*Figure 6C*), suggesting that the increased digestion ability in *nsy-1* mutant animals is mainly dependent on the intestinal digestion factor BCF-1. This regulatory cascade highlights the intricate link between neuronal signals and gut responses, ensuring an adaptive reaction to harmful food. We speculated that except for INS-23 there should be other factors as signaling regulated by neuronal NSY-1 to inhibit intestinal digestion factor BCF-1 for digestion shutdown.

Future studies should focus on delineating the precise molecular pathways linking NSY-1 signaling in AWC neurons to digestion regulation in the gut. Identifying the role of other sensory neurons in food quality assessment and their interactions with the digestive system may uncover new facets of neuron-gut communication and adaptive responses.

In summary, our study reveals a sophisticated mechanism in *C. elegans* that integrates neuronal detection of harmful food sources with systemic digestive responses. This neuron-digestive crosstalk is crucial for maintaining organismal homeostasis and survival in the presence of toxic food sources. The findings suggest that similar pathways may exist in other species, including humans, providing a foundation for future research on food safety, toxin avoidance, and the neural regulation of digestive processes. This has important implications for public health, as it illuminates the biological mechanisms underlying foodborne illness and the body's defenses against such threats.

## Materials and methods

### *C. elegans* strains and maintenance

Nematode stocks were maintained on nematode growth medium (NGM) plates seeded with bacteria (*E. coli* OP50) at 20°C.

The following strains/alleles were obtained from the Caenorhabditis Genetics Center (CGC) or as indicated:

1. The following strains were obtained from CGC:
   N2 Bristol (wild-type control strain);
   AU3: *nsy-1(ag3)*;
   ZD101: *tir-1(qd4)*;
   RB1971: *bcf-1(ok2599)*;

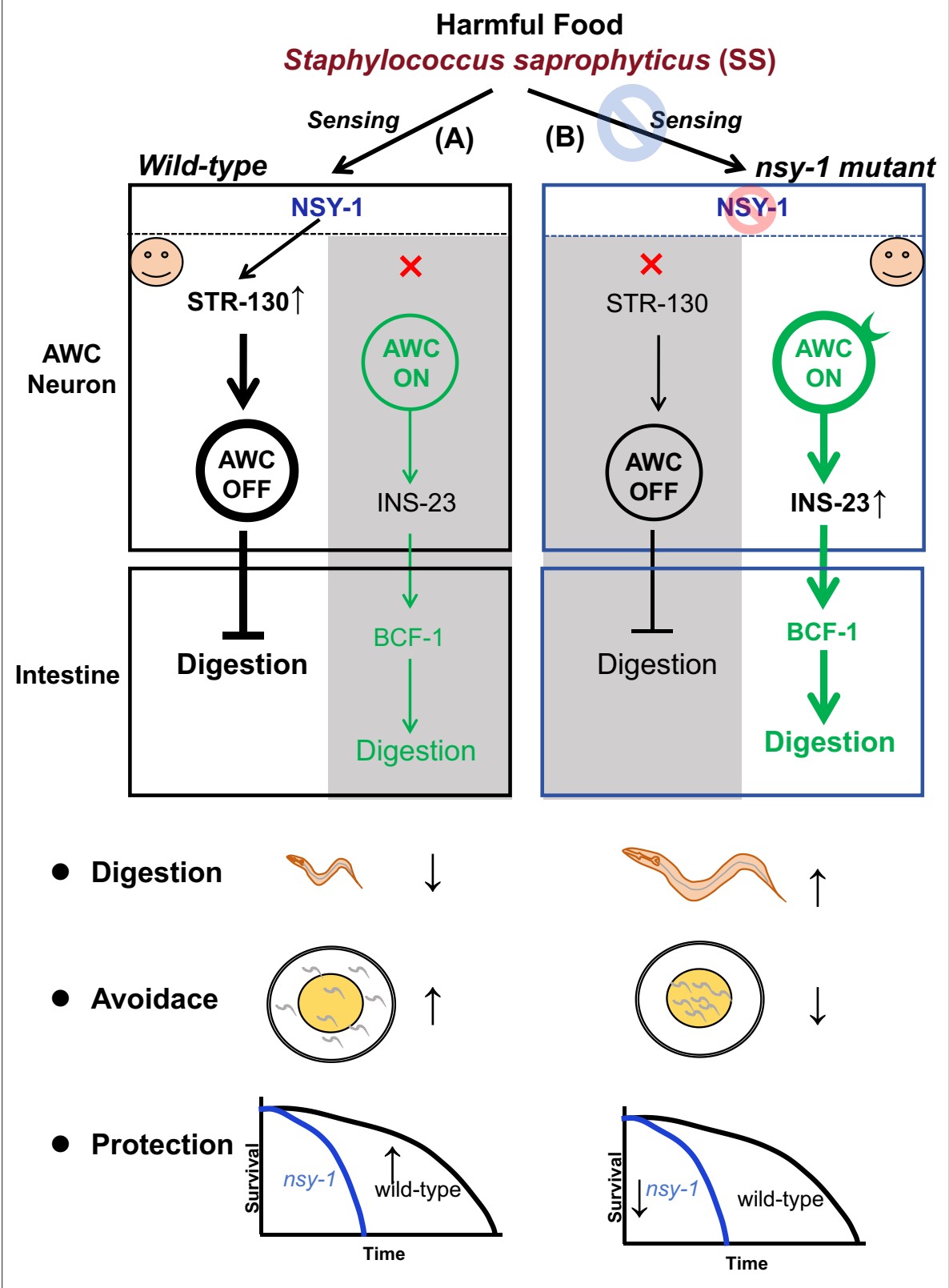

**Figure 7.** A model reveals a neural-digestive mechanism for evaluating harmful food. (**A**) AWC neuron-expressed NSY-1 detects *Staphylococcus saprophyticus* (SS) as harmful food and shuts down digestion by inducing the AWC^OFF neural circuit and NSY-1-dependent STR-130. This mechanism protects animals and helps them avoid harmful food. (**B**) Mutations in NSY-1 lead to SS digestion by activating the insulin/IGF-1 signaling (IIS) pathway and BCF-1 expression, thereby reducing the animals' ability to avoid harmful food and decreasing their protection.

KU25: *pmk-1(km25)*;
KU4: *sek-1(km4)*;
PY7502: *oyIs85[ceh-36p::TU#813+ceh-36p::TU#814+srtx-1p::GFP+unc-122p::DsRed]*
CB1370: *daf-2(e1370)*; shared from Mintie Pu lab
CX3695: *str-2::gfp+lin-15(+)*; shared from Huanhu Zhu lab
CX4998: *str-2::gfp+lin-15(+);nsy-1(ky397)*; shared from Hongyun Tang lab

2. The following strains were obtained from published papers:
PHX4067: [*Pbcf-1::bcf-1::gfp::3xflag*] (*He et al., 2023*);

3. The following strains were constructed by this study:
YNU186: *ylfEx124[Pnsy-1::gfp;Podr-1::rfp]* was constructed by injecting plasmid *Pnsy-1::nsy-1::gfp* with *Podr-1::rfp* in N2 background;
YNU189: *Pbcf-1::bcf-1::gfp::3xflag;nsy-1(ag3)* was constructed by crossing PHX4067[*Pbcf-1::bcf-1::gfp::3xflag*] with AU3[*nsy-1(ag3)*];
YNU238: *ylf6(nsy-1*, EMS mutant);
YNU465: *ylfEx252[Podr-1::nsy-1::gfp;Podr-1::rfp;nsy-1(ag3)]* was constructed by injecting plasmid *Podr-1::nsy-1::gfp* with *Podr-1::rfp* in *nsy-1(ag3)* background;
YNU466: *ylfEx253[Pvha-6::nsy-1::gfp;Podr-1::rfp;nsy-1(ag3)]* was constructed by injecting plasmid *Pvha-6::nsy-1::gfp* with *Podr-1::rfp* in *nsy-1(ag3)* background;
YNU488: *ylfEx259[Pstr-130::str-130::mcherry;Podr-1::rfp;nsy-1(ag3)]* was constructed by injecting plasmid *Pstr-130::str-130::mcherry* with *Podr-1::rfp* in *nsy-1(ag3)* background;
YNU491: *ylf57*, AWC neuron specific knock out *nsy-1* strain was constructed by injecting plasmid pDD162[P*odr-1::Cas9*+P*u6::nsy-1-sg*], *nsy-1* repair template (synthesis from Tsingke), pDD162[P*eft-3::Cas9*+P*u6::dpy-10-sg*], dpy-10 repair template(synthesis from Tsingke) in PHX4067(P*bcf-1::bcf-1::gfp::3xflag*) background;
YNU501: *nsy-1(ag3);bcf-1(ok2599)* double mutant was constructed by crossing RB1971[*bcf-1(ok2599)*] with AU3[*nsy-1(ag3)*];
YNU508: *ylfEx266[Pstr130::str-130::mcherry;Podr-1::rfp;nsy-1(ag3)]* was constructed by injecting plasmid *Pstr130::str-130::mcherry* with *Podr-1::rfp* in *str-2::gfp+lin-15(+)* background;
YNU517: *nsy-1(ag3);daf-2(e1370)* double mutant was constructed by crossing CB1370[*daf-2(e1370)*] with AU3[*nsy-1(ag3)*];
YNU729: *ylfEx351[Pges-1::daf-2::gfp;Podr-1::rfp;nsy-1(ag3);daf-2(e1370)]* was constructed by injecting plasmid *Pges-1::daf-2::gfp* with *Podr-1::rfp* in *nsy-1(ag3);daf-2(e1370)* background;
YNU732: *ylfIs48[Podr-1::nsy-1::gfp;Podr-1::rfp;nsy-1(ag3)]* was constructed by injecting plasmid *Podr-1::nsy-1::gfp* with *Podr-1::rfp* in *nsy-1(ag3)* background;
YNU730: *ylfEx352[Pceh-36::sid-1::mcherry;pRF4(rol-6);Pbcf-1::bcf-1::gfp::3xflag;sid-1(qt9)]* was constructed by injecting plasmid *Pceh-36::sid-1::mcherry* with *pRF4(rol-6)* in *Pbcf-1::bcf-1::gfp::3xflag;sid-1(qt9)* background;
YNU731: *ylfEx353[Pins-23::ins-23::gfp;Podr-1::rfp]* was constructed by injecting plasmid *Pins-23::ins-23::gfp* with *Podr-1::rfp* in N2 background.

## Bacteria strains

*E. coli*-OP50 (from CGC) and SS (from ATCC) were cultured at 37°C in LB medium. A standard overnight cultured bacteria was then spread onto each Nematode growth media (NGM) plate.

## Generation of transgenic strains

1. To construct the *C. elegans* plasmid for expression of *nsy-1*, 1527 bp promoter of *nsy-1* was inserted into the pPD95_77-gfp vector. DNA plasmid mixture containing P*nsy-1::GFP* (20 ng/ul) and P*odr-1p::RFP*(50 ng/ul) was injected into the gonads of adult wild-type N2 animals.

2. To construct the *C. elegans* plasmid for expression of *nsy-1* in AWC neuron, 1348 bp promoter of *odr-1* and genomic DNA of *nsy-1* was inserted into the pPD49.26-gfp vector. DNA plasmid mixture containing P*odr-1::nsy-1::GFP* (20 ng/ul) and P*odr-1::RFP* (50 ng/ul) was injected into the gonads of adult *nsy-1(ag3)*.

3. To construct the *C. elegans* plasmid for expression of *str-130*, 2000 bp promoter of *str-130* and 1324 bp genomic DNA of *str-130* was inserted into the pPD49.26-mcherry vector. DNA plasmid mixture containing P*str-130::str-130::mcherry* (20 ng/μl) and P*odr-1::rfp* (50 ng/μl) was injected into the gonads of adult CX3695[*str-2::gfp+lin-15(+)*].

4. To construct the *C. elegans* plasmid for expression of *daf-2* in intestine, 2549 bp promoter of *ges-1* and 5400 bp cDNA of *daf-2* was inserted into the pPD95_77-gfp vector. DNA plasmid mixture containing P*ges-1::daf-2::gfp*(20 ng/µl) and P*odr-1::rfp* (50 ng/µl) was injected into the gonads of adult *nsy-1(ag3);daf-2(e1370)*.

5. To construct the *C. elegans* plasmid for expression of *sid-1* in AWC neuron, 2000 bp promoter of *ceh-36* and 2328 bp cDNA of *sid-1* was inserted into the pPD49.26-mcherry vector. DNA plasmid mixture containing P*ceh-36::sid-1::mcherry* (20 ng/µl) and *rol-6* (50 ng/µl) was injected into the gonads of adult *Pbcf-1::bcf-1::gfp::3xflag;sid-1(qt9)*.

6. To construct the *C. elegans* plasmid for expression of *ins-23*, 2017 bp promoter of *ins-23* and 286 bp genomic DNA of *ins-23* was inserted into the pPD49.26-gfp vector. DNA plasmid mixture containing P*ins-23::ins-23::gfp* (20 ng/µl) and P*odr-1::rfp* (50 ng/µl) was injected into the gonads of adult wild-type N2 animals.

## Generation *nsy-1* AWC neuron-specific knockout strain and genotyping

To construct the *C. elegans* plasmid for knockout of *nsy-1* in AWC neuron, 600 bp promoter of *eft-3* was replaced by 1348 bp promoter of *odr-1* and *nsy-1* sgRNA was also inserted into the same CRISPR-Cas9-sgRNA vector pDD162 (*Dickinson et al., 2013*). The Cas9 target sites were designed via CRISPR design tool (http://crispor.tefor.net/) and the sgRNA sequence was 5'-GAATTTACGCGTTCGAGAAA TGG-3'. Knockout strains were generated by injecting 25 ng/µl Cas9-sgRNA plasmid, 2 µM repair template, co-injection markers include 20 ng/µl *dpy-10* Cas9-sgRNA plasmid and 2 µM *dpy-10* repair template.

Worms were picked into 10 µl of worm lysis buffer (50 mM KCl, 10 mM Tris–HCl pH 8.0, 2.5 mM MgCl$_2$, 0.45% NP40, 0.45% Tween-20, 0.01% gelactin, 0.2 mg/ml proteinase K), quickly freeze–thaw three times using liquid nitrogen, incubated it at 60°C for 90 min and 95°C for 20 min. 1 µl supernatant was taken and performed for PCR analysis with the following primers: *nsy-1*: forward 5'-CAAGAGGC AAGTGCAGCATA-3', reverse 5'-TGACTGTCCCATGCTCTCAC-3', then digested with NheI endonuclease overnight and identified by DNA agarose electrophoresis.

## Preparation of SS

SS preparation was followed by our published protocol (*Geng et al., 2022*; *Liu and Qi, 2023*). Briefly, a standard overnight culture of SS (37°C in LB broth) was diluted into fresh LB broth (1:100 ratio). SS was then spread onto each NGM plate when the diluted bacteria grew to OD600=0.5.

## Analysis of worm's growth in SS bacteria

The standard overnight cultured SS was then spread onto 60 mm NGM plate. Worms were grown on *E. coli* - OP50, and eggs were collected by bleaching and then washing in M9 buffer. Synchronized L1 larvae were obtained by allowing the eggs to hatch in M9 buffer for 12 h. Synchronized L1s were seeded to plates prepared for the specific assay and incubated at 20°C for 4 days. The developmental conditions were determined by body length.

## Food behavior assay

### Food avoidance assay

Food avoidance assay was performed following our published methods (*Qi et al., 2017*). Briefly, 5 ul of overnight cultured bacteria was seeded on the center of 60 mm NGM plates. About 30–50 synchronized L1 animals were seeded onto the bacterial lawns and cultured at 20°C for 8 h. The aversion index was determined by N(out of lawn)/N(total).

### Food choice assay

Food choice assay was performed following our published methods (*Qi et al., 2017*). Briefly, 5 ul of overnight cultured bacteria was seeded on the different side of 60 mm NGM plates. About 300 synchronized L1 animals were seeded onto the center of NGM plates and cultured at 20°C for 8 h. The food trend index was determined by N (selecting food1)/[N(selecting food1)+N(selecting food2)] and N(selecting food2)/[N(selecting food1)+N(selecting food2)].

## Lifespan analysis

### Larval lifespan

Studies of larval lifespan were performed on NGM plates at 20°C as previously described (*Cui et al., 2013*). Briefly, L1 staged worms were placed on NGM plates or SS-seeded NGM plates. Worms were scored every day. Prism8 software was used for statistical analysis.

### Adult lifespan

Studies of lifespan were performed on NGM plates at 20°C as previously described (*Kimura et al., 1997*).

Briefly, lifespan was begun on day 0 by placing healthy L4 stage hermaphrodites onto OP50 seeded NGM plates. Animals were transferred to a fresh OP50 or SS seeded plates during the reproductive period (approximately the first 10 days) to eliminate self-progeny and every 2 days thereafter. Worms were scored every day. Prism8 software was used for statistical analysis.

## Ethyl methanesulfonate (EMS)-induced mutagenesis

Synchronized L1 animals were grown to the L4 stage on OP50 and then subjected to a 4 h treatment with 0.5% ethyl methanesulfonate (EMS). After treatment, the P0 generation animals were thoroughly washed with M9 solution and transferred to OP50 plates to grow and produce the F1 generation. Groups of 3–4 F1 animals were picked and placed onto new OP50 plates (3–4 F1 worms per plate) to generate the F2 generation through self-fertilization. Once the F2 generation reached adulthood, all the F2 animals were bleached to obtain the next generation of eggs. Synchronized L1 larvae were obtained by allowing the eggs to hatch in M9 buffer for 12 h. The synchronized L1s were then seeded onto SS plates and incubated at 20°C for 4 days. Candidate mutants that could grow on SS plates were identified. To confirm the suspected mutants, each candidate mutant was picked singly onto OP50 for passaging and growth to adulthood. These adult animals were then bleached to obtain synchronized L1 mutants. The synchronized L1 mutants were seeded onto SS plates to confirm their ability to grow on SS.

## Identification of EMS mutants

DNA isolation, library construction, and whole-genome sequencing (WGS) with gene identification were carried out according to the published protocol (*Joseph et al., 2018*).

### Preparation of samples for WGS

For each backcross, wild-type males were crossed to mutant hermaphrodites, F1 cross-progeny animals were individually cloned and F2 variants and wild-types were isolated under SS feeding condition. The variant strains and wild-type strains were mixed respectively to constitute the 'DNA-pool' used as samples for WGS.

WGS of pooled F2 recombinants, homozygous for the mutant phenotype following two outcrosses to wild-type N2 animals, was performed to identify the mutations.

### WGS data processing

For WGS, paired-end libraries were sequenced on an Illumina HiSeq 2000. Fastqc was used to control the quality of raw data, and Trimmomatic was used to filter the data. Bwa was used to construct the index of *C. elegans* genome and align clean reads to the reference gene sequence (Species: *Caenorhabditis_elegans*; source: UCSC; reference genome version: WBcel235/ce11). Samtools was used for file format conversion and sorting. Picard was used to remove the duplicate reads and then GATK was used to identify the intervals and realign. The realigned sequence was piled up by Samtools and then inputted to Varscan to call the variants including SNPs and INDELs. Vcflib was used to perform subtraction between the wild-types and variants. The file was annotated by Snpeff, and the candidate genes were finally obtained.

## Preparation of samples for RNA sequencing

RNA-seq was done with three biological replicates that were independently generated, collected, and processed. Adult wild-type (N2) or *nsy-1* mutant worms were bleached and then the eggs were

incubated in M9 for 18 h to obtain synchronized L1 worms. Synchronized L1 worms were cultured in the NGM plate seeded with OP50 or SS for 4 h at 20°C. L1 worms were then collected for sequencing.

### Fluorescence microscopy of *C. elegans*

Slides for imaging were prepared by making a fresh flattened 5% agarose pad. Worms were mounted on 5% agar pads in M9 buffer with 5 mM levamisole then sealed beneath a 22 × 22 mm coverglass. Imaging was done using an Olympus BX53 microscope with a DP80 camera. *str-2::gfp* or *bcf-1::gfp* expression was measured through imaging.

### Microscopy

The fluorescence photographs were taken using an Olympus BX53 microscope with a DP80 camera. Development statistics were taken using Olympus MVX10 dissecting microscope with a DP80 camera.

### Quantification and statistical analysis

Quantification

Animals were randomly selected for fluorescent photography. The size of transgene worms was photographed using the Nomarski microscope and measured using ImageJ software. ImageJ software was used for quantifying fluorescence intensity of indicated animals, which was then normalized with the control group.

Statistical analysis

All experiments were performed independently at least three times with similar results. All statistical analyses were performed using the unpaired two-tailed Student's test. Statistical parameters are presented as mean ± SD, statistical significance (*p<0.05, **p<0.01, ***p<0.001), and 'n' (the number of worms counted).

The log-rank (Mantel–Cox) test was used for lifespan assay, and the exact p values of statistics for all survival assays are listed in the figures.

## Acknowledgements

We thank the Caenorhabditis Genetics Center (CGC) (funded by NIH P40OD010440) for strains. We thank Dr. Huanhu Zhu and Dr. Xiajing Tong (ShanghaiTech University), Dr. Mintie Pu (Yunnan University), Dr. Hongyun Tang (Westlake University), and Dr. Zhiyong Shao (Fudan University) for sharing strains. This work was supported by the Yunnan Provincial Science and Technology Project at Southwest United Graduate School (202302AP370005 to BQ), Yunnan Provincial Science and Technology Project (202201AT070196 to BQ), Yunnan Revitalization Talent Support Program (C619300A086 to ZS, K264202230211 to BQ), Ministry of Science and Technology of the People's Republic of China (2019YFA0803100, 2019YFA0802100 to BQ), and the National Natural Science Foundation of China (32071129 to ZS, 32170794 to BQ).

## Additional information

### Funding

| Funder | Grant reference number | Author |
| --- | --- | --- |
| Yunnan Provincial Science and Technology Department | 202302AP370005 | Bin Qi |
| Ministry of Science and Technology of the People's Republic of China | 2019YFA0803100 | Bin Qi |
| National Natural Science Foundation of China | 32071129 | Zhao Shan |

| Funder | Grant reference number | Author |
| --- | --- | --- |
| National Natural Science Foundation of China | 32170794 | Bin Qi |
| Yunnan Revitalization Talent Support Program | C619300A086 | Zhao Shan |
| Yunnan Revitalization Talent Support Program | K264202230211 | Bin Qi |
| Yunnan Provincial Science and Technology Department | 202201AT070196 | Bin Qi |
| Ministry of Science and Technology of the People's Republic of China | 2019YFA0802100 | Bin Qi |

The funders had no role in study design, data collection and interpretation, or the decision to submit the work for publication.

## Author contributions

Yating Liu, Data curation, Formal analysis, Validation, Investigation, Visualization, Methodology, Project administration, Writing – review and editing; Guojing Tian, Data curation, Formal analysis, Validation, Investigation, Visualization, Methodology, Writing – original draft, Project administration; Ziyi Wang, Junkang Zheng, Huimin Liu, Data curation, Investigation, Methodology; Sucheng Zhu, Supervision, Methodology; Zhao Shan, Conceptualization, Supervision, Funding acquisition, Writing – original draft, Project administration, Writing – review and editing; Bin Qi, Conceptualization, Resources, Supervision, Funding acquisition, Validation, Visualization, Writing – original draft, Project administration, Writing – review and editing

## Author ORCIDs

Zhao Shan ⓘ https://orcid.org/0000-0001-5064-1023
Bin Qi ⓘ https://orcid.org/0000-0003-2261-1550

Reviewer #1 (Public review): https://doi.org/10.7554/eLife.104028.3.sa1
Reviewer #2 (Public review): https://doi.org/10.7554/eLife.104028.3.sa2
Reviewer #3 (Public review): https://doi.org/10.7554/eLife.104028.3.sa3
Author response https://doi.org/10.7554/eLife.104028.3.sa4

# Additional files

## Supplementary files

Supplementary file 1. List of genes induced by SS food that are dependent on NSY-1. Related to *Figure 4B*.

Supplementary file 2. GO enrichment analysis of 304 NSY-1-dependent candidate genes responding to SS. Related to *Figure 4—figure supplement 1A and B*.

Supplementary file 3. List of genes induced in *nsy-1(ag3)* mutant animals feeding on SS. Related to *Figure 5B*.

Supplementary file 4. GO enrichment analysis of 308 genes induced by the nsy-1 mutation under SS feeding conditions. Related to *Figure 5—figure supplement 1A and B*.

MDAR checklist

## Data availability

Sequencing data have been deposited in CNCB under accession codes PRJCA042026. All data generated or analyzed during this study are included in the manuscript and supporting files; source data files have been provided for all Figures. This paper does not report original code. Any additional information required to reanalyze the data reported in this paper is available from the lead contact upon request (qb@ynu.edu.cn).

The following dataset was generated:

| Author(s) | Year | Dataset title | Dataset URL | Database and Identifier |
|---|---|---|---|---|
| Liu Y, Qi B | 2025 | RNA-seq data of different food-2 | https://ngdc.cncb.ac.cn/bioproject/browse/PRJCA042026 | China National Center for Bioinformation, PRJCA042026 |

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
