## [Editor Report · eLife Assessment]

This **important** study investigates how signals from the nervous system can influence the response to different food sources. To demonstrate the role of specific neuronal and intestinal regulators in sensing food quality and modulating digestion, the authors present evidence through a combination of genetic screening, RNA-seq analysis, and functional studies. These findings shed light on an adaptive strategy to integrate food perception with physiological responses, with a mix of **solid** and **convincing** evidence supporting the work.

---

## [Referee Report · Reviewer #1 (Public review)]

Summary:

In this manuscript, Liu et al have tried to dissect the neural and molecular mechanisms that *C. elegans* use to avoid the digestion of harmful bacterial food. Liu et al show that *C. elegans* use ON-OFF state of AWC olfactory neurons to regulate the digestion of harmful gram-positive bacteria S. saprophyticus (SS). Authors show that when *C. elegans* are fed on SS food, AWC neurons switch to OFF fate, which prevents the digestion of S. saprophyticus, and this helps *C. elegans* avoid these harmful bacteria. Using genetic and transcriptional analysis as well as making use of previously published findings, Liu et al implicate p38 MAPK pathway (in particular, NSY-1, the *C. elegans* homolog of MAPKKK ASK1) and insulin signaling in this process.

Strengths:

The revised manuscript has improved significantly. The authors have addressed almost all the comments that I had in my initial review.

Weaknesses:

None.

---

## [Referee Report · Reviewer #2 (Public review)]

Summary:

Using *C. elegans* as a model, the authors present an interesting story demonstrating a new regulatory connection between olfactory neurons and the digestive system. Mechanistically, they identified key factors (NSY-1, STR-130 et.al) in neurons, as well as critical 'signaling factors' (INS-23, DAF-2) that bridge different cells/tissues to execute the digestive shutdown induced by poor-quality food (Staphylococcus saprophyticus, SS).

Strengths:

The conclusions of this manuscript are mostly well supported by the experimental results shown.

Weaknesses:

The authors have done a nice job in addressing my comments.

---

## [Referee Report · Reviewer #3 (Public review)]

Summary:

The study explores a molecular mechanism by which *C. elegans* detects low-quality food through neuron-digestive crosstalk, offering new insights into food quality control systems. Liu and colleagues demonstrated that NSY-1, expressed in AWC neurons, is a key regulator for sensing Staphylococcus saprophyticus (SS), inducing avoidance behavior and shutting down the digestive system via intestinal BCF-1. They further revealed that INS-23, an insulin peptide, interacts with the DAF-2 receptor in the gut to modulate SS digestion. The study uncovers a food quality control system connecting neural and intestinal responses, enabling *C. elegans* to adapt to environmental challenges.

Strengths:

The study employs a genetic screening approach to identify nsy-1 as a critical regulator in detecting food quality and initiating adaptive responses in *C. elegans*. The use of RNA-seq analysis is particularly noteworthy, as it reveals distinct regulatory pathways involved in food sensing (Figure 4) and digestion of Staphylococcus saprophyticus (Figure 5). The strategic application of both positive and negative data mining enhances the depth of analysis. Importantly, the discovery that *C. elegans* halts digestion in response to harmful food and employs avoidance behavior highlights a physiological adaptation mechanism.

Weaknesses:

Major weaknesses have been addressed.

---

## [Author Response]

The following is the authors’ response to the original reviews.

**Reviewer #1 (Public review):**
Summary:In this manuscript, Liu et al have tried to dissect the neural and molecular mechanisms that *C. elegans* use to avoid digestion of harmful bacterial food. Liu et al show that *C. elegans* use the ON-OFF state of AWC olfactory neurons to regulate the digestion of harmful gram-positive bacteria S. saprophyticus (SS). The authors show that when *C. elegans* are fed on SS food, AWC neurons switch to OFF fate which prevents digestion of S. saprophyticus and this helps *C. elegans* avoid these harmful bacteria. Using genetic and transcriptional analysis as well as making use of previously published findings, Liu et al implicate the p38 MAPK pathway (in particular, NSY-1, the *C. elegans* homolog of MAPKKK ASK1) and insulin signaling in this process.Strengths:The authors have used multiple approaches to test the hypothesis that they present in this manuscript.Weaknesses:Overall, I am not convinced that the authors have provided sufficient evidence to support the various components of their hypothesis. While they present data that loosely align with their hypothesis, they fail to consider alternative explanations and do not use rigorous approaches to strengthen their overall hypothesis. The selective picking of genes from the RNA sequencing data and forcing the data to fit the proposed hypothesis based on previously published findings, without exploring other approaches, indicates a lack of thoroughness and rigor. These critical shortcomings significantly diminish enthusiasm for the manuscript in its totality. In my opinion, this is the biggest weakness in this manuscript.

We appreciate the reviewer’s all the suggestions which help us to improve this paper. We now addressed reviewer’s comments at the section of “Reviewer #1 (Recommendations for the authors)”

**Reviewer #2 (Public review):**
Summary:Using *C. elegans* as a model, the authors present an interesting story demonstrating a new regulatory connection between olfactory neurons and the digestive system.Mechanistically, they identified key factors (NSY-1, STR-130 et.al) in neurons, as well as critical 'signaling factors' (INS-23, DAF-2) that bridge different cells/tissues to execute the digestive shutdown induced by poor-quality food (Staphylococcus saprophyticus, SS).Strengths:The conclusions of this manuscript are mostly well supported by the experimental results shown.Weaknesses:Several issues could be addressed and clarified to strengthen their conclusions.(1) The word "olfactory" should be carefully used and checked in this manuscript. Although AWCs are classic olfactory neurons in *C. elegans*, no data in this manuscript supports the idea that olfactory signals from SS drive the responses in the digestive system. To validate that it is truly olfaction, the authors may want to check the responses of worms (e.g. AWC, digestive shutdown, INS-23 expression) to odors from SS.

We appreciate the reviewer’s careful attention to terminology. We agree that the term "olfactory" requires direct experimental validation. However, in this paper, we only used "olfactory" to specific define the AWC neurons. As reviewer’s suggestion, we now deleted the word “olfactory”.

(2) In line 113, what does "once the digestive system is activated" mean? The authors need to provide a clearer statement about 'digestive activation' and 'digestive shutdown'.

Previously, we observed that activating larval digestion with heat-killed *E. coli* or *E. coli* cell wall peptidoglycan (PGN) enabled the digestion of SS as food (Hao et al., 2024). Additionally, when animals reached the L2 stage by feeding normal OP50 diet, they could utilize SS as a food source to support growth (Figure 1**—**figure supplement 1D). These findings suggest that once digestion is activated (via *E. coli* components or L2-stage maturation), worms gain the capacity to process SS as a viable food source, abolishing SS-induced growth impairment (Hao et al., 2024) (Figure 1**—**figure supplement 1D).

(3) No control data on OP50. This would affect the conclusions generated from Figures 2A, 2B, 2D, 3B, 3C, 3G, 4D-G, 5D-E, 6B-D.

We appreciate this point. The central goal of the experiments listed (Figures 2A,B,D; 3B,C,G; 4D-G; 5D-E; 6B-D) was not to compare growth or behavior between SS and OP50 under standard conditions, but rather to understand the genetic basis of the *C. elegans* response specifically to SS, as identified through our nsy-1 mutant screen.

Our data in Figure 1 clearly establishes the fundamental difference in growth and feeding behavior when larvae encounter SS compared to OP50 (Figures 1A,B). Having established SS as an unfavorable food source that triggers a specific protective response (digestive shutdown), the subsequent experiments focus on deciphering how this response is mediated.

Therefore, within these specific experimental contexts under SS feeding: The primary comparison is between wild-type (N2) and nsy-1 mutant animals. All assays (growth, behavior, survival) are performed under the same SS feeding conditionsfor both genotypes.

This design allows us to directly assess the functional role of NSY-1 in mediating the SS-specific response pathway we are investigating. Including an OP50 control for every figure would not address this core genetic question and could introduce confounding variables given the established difference in how *C. elegans* treats these two food sources. The critical internal control for these specific experiments is the performance of the wild-type under SS versus the mutant under SS.

(4) Do the authors know which factors are released from AWC neurons to drive the digestive shutdown?

Enrichment analysis revealed that genes related to extracellular functions, such as insulin-related genes, are induced in nsy-1 mutant animals (Figure 5—figure supplement 1A, Supplementary file 4). Further analysis of insulin-related genes from the RNA-seq data showed that ins-23 is predominantly induced in nsy-1 mutant animals (Figure 5—figure supplement 1B), suggesting its potential role in promoting SS digestion. We found that knockdown of ins-23 in nsy-1 mutants inhibited SS digestion (Figure 5D). Given that INS-23 is expressed in AWC neurons (Figure 5**—**figure supplement 3A, CeNGEN), this suggests increased production and likely enhanced release of INS-23 from AWC neurons in the nsy-1 mutant background, which promotes SS digestion.

The insulin/insulin-like growth factor signaling (IIS) pathway, particularly through the DAF-2 receptor, integrates nutritional signals to regulate various behavioral and physiological responses related to food (Kodama et al., 2006; Ryu et al., 2018). It has been shown that INS-23 acts as an antagonist for the DAF-2 receptor to promote larval diapause (Matsunaga et al., 2018). To test whether ins-23 induction in nsy-1 mutants promotes SS digestion through its receptor, DAF-2, we constructed a nsy-1; daf-2 double mutant. We found that the SS digestion ability of the nsy-1 mutant was inhibited by the daf-2 mutation. This suggests that the nsy-1 mutation induces the insulin peptide ins-23, which promotes SS digestion through its potential receptor, DAF-2.

The data supports a model where AWC neurons regulate digestion via the release of INS-23. Loss of nsy-1 function increases INS-23 release from AWC, activating DAF-2 signaling and promoting digestion. Conversely, in wild-type animals, reduced INS-23 release from AWC contributes to digestive shutdown in response to SS food.

**Reviewer #3 (Public review):**
Summary:The study explores a molecular mechanism by which *C. elegans* detects low-quality food through neuron-digestive crosstalk, offering new insights into food quality control systems. Liu and colleagues demonstrated that NSY-1, expressed in AWC neurons, is a key regulator for sensing Staphylococcus saprophyticus (SS), inducing avoidance behavior and shutting down the digestive system via intestinal BCF-1. They further revealed that INS-23, an insulin peptide, interacts with the DAF-2 receptor in the gut to modulate SS digestion. The study uncovers a food quality control system connecting neural and intestinal responses, enabling *C. elegans* to adapt to environmental challenges.Strengths:The study employs a genetic screening approach to identify nsy-1 as a critical regulator in detecting food quality and initiating adaptive responses in *C. elegans*. The use of RNA-seq analysis is particularly noteworthy, as it reveals distinct regulatory pathways involved in food sensing (Figure 4) and digestion of Staphylococcus saprophyticus (Figure 5). The strategic application of both positive and negative data mining enhances the depth of analysis. Importantly, the discovery that *C. elegans* halts digestion in response to harmful food and employs avoidance behavior highlights a physiological adaptation mechanism.Weaknesses:Major points:(1) While NSY-1 positively regulates str-130 expression in AWC neurons and is critical for SS avoidance and survival, the authors should examine whether similar phenotypes are observed in str-130 mutants.

In this study, we mainly focused on how worms sense adverse food sources (SS food) and shutdown digestion (not growth as digestion shutdown readout). We found that nsy-1 in AWC play key roles in response SS food, once nsy-1 mutation, mutant animals cannot detect SS food and digest it, therefore growth under SS food. From RNA-seq, we found that nsy-1 positively regulates several sensory perception related genes (sra-32, str-87, str-112, str-130, str-160, str-230) (Figure 4**—**figure supplement 1A, Supplementary file 2). After screen, we found that we found that knockdown of str-130 in wild-type animals promoted SS digestion, thereby supporting animal growth (Figure 4D), and the proportion of animals with two AWC^OFF^ neurons decreased (Figure 4E). Secondly, we found that overexpression of str-130 in nsy-1 mutant animals inhibited SS digestion, thereby slowing animal growth (Figure 4F), and the proportion of animals with two AWC^OFF^ neurons increased (Figure 4G). These results demonstrate that NSY-1 promotes the AWC^OFF^ state by inducing str-130 expression, which in turn inhibits SS digestion in *C. elegans*.

(2) NSY-1 promotes the AWC-OFF state through str-130, inhibiting SS digestion. The authors should investigate whether STR-130 in AWC neurons regulates bcf-1 expression levels in the intestine.

We agree with the reviewer's suggestion regarding the potential role of STR-130 in AWC neurons regulating intestinal bcf-1 expression. To address this, we generated transgenic worms with AWC-specific knockdown of str-130, achieved by rescuing sid-1 cDNA expression under the ceh-36 promoter (AWC-specific) in sid-1(qt9);BCF-1::GFP background worms.

We observed that AWC neuron-specific RNAi of str-130 elevated intestinal BCF-1::GFP expression (Figure 6—figure supplement 1B). This demonstrates that STR-130 functions cell-non-autonomously in AWC neurons to repress BCF-1 expression in the intestine.

(3) The current results rely on str-2 expression levels to indicate the AWC state. Ablating AWC neurons and testing the effects on digestion would provide stronger evidence for their role in digestive regulation.

To confirm the important of AWC state in SS digestion, we performed AWC-specific neuron ablation experiments using previously validated transgenic strain that expresses cleaved caspase under the AWC-specific promoter, ceh-36 (ceh-36p::caspase). Critically, worms with ablated AWC neurons completely failed to digest SS food (Figure 3—figure supplement 4), phenocopying the non-digesting state of wild-type worms on SS when AWC-OFF signaling is impaired. This result directly confirms that functional AWC neurons are essential for initiating SS digestion, aligning with our model where the AWC-OFF state (induced by SS) inhibits digestion while the AWC-ON state promotes it.

Furthermore, we previously study discovered that AWC ablation activates the intestinal mitochondrial unfolded protein response and inhibits food digestion, mechanistically linking neuronal integrity to gut stress responses and digestive inhibition.

Together, these functional ablation studies provide compelling physiological evidence that AWC neurons act as central regulators of food-state sensing and gut function.

(4) The claim that NSY-1 inhibits INS-23 and that INS-23 interacts with DAF-2 to regulate bcf-1 expression (Line 339-340) requires further validation. Neuron-specific disruption of INS-23 and gut-specific rescue of DAF-2 should be tested.

We agree with the reviewer that the proposed NSY-1 ⊣ INS-23 → DAF-2 → BCF-1 signaling axis requires tissue-specific validation. To address this, we conducted compartment-specific functional dissection of INS-23 and DAF-2:

AWC neuronal role of INS-23:

To test whether INS-23 acts in AWC neurons to regulate intestinal BCF-1, we generated AWC-specific knockdown strains which was achieved by rescuing sid-1 cDNA expression under the ceh-36 promoter in a sid-1(qt9);BCF-1::GFP background. We found that AWC-restricted ins-23 knockdown significantly reduced intestinal BCF-1::GFP expression (Figure 6—figure supplement 1A). This confirms that INS-23 functions cell-non-autonomously within AWC sensory neurons to activate intestinal BCF-1, consistent with NSY-1’s upstream inhibition of INS-23 in this neuronal subtype

Intestinal role of DAF-2 as INS-23 receptor:

To investigate weather DAF-2 acts as the gut-localized receptor for neuronal INS-23 signaling, we performed tissue-specific rescue experiments in the nsy-1(ag3);daf-2(e1370) double mutant. When DAF-2 was re-introduced specifically in the intestine (using the ges-1 promoter), we observed a significant suppression of SS digestion (Figure 5—figure supplement 3B), but not rescue digestive defect. This indicates that INS-23 induction in nsy-1 mutants promotes digestion independently of intestinal DAF-2 function.

(5) Figure Reference Errors: Lines 296-297 mention Figure 6E, which does not exist in the main text. This appears to refer to Figure 5E, which has not been described.

We corrected this.

**Reviewer #1 (Recommendations for the authors):**
I would like the authors to address the following comments in a resubmission.(1) The hallmark of the activated p38 MAPK pathway is the phosphorylation of most downstream kinase p38 (PMK-1/PMK2 in *C. elegans*) of this kinase cascade. Previous work from Bergmann lab showed that the most downstream kinase of this pathway, PMK-1/PMK-2, is not required for AWC asymmetry. I wonder whether that is the case also for the model that Liu et al have presented in this manuscript. Since p38/PMK-1 undergoes activation (phosphorylation) in response to pathogenic bacteria like *P. aeruginosa*, it is worth testing whether PMK-1 plays a role downstream of NSY-1 in the model that Liu et al present in this manuscript. It would be worth testing whether there is increased phosphorylation of p38 when *C. elegans* are fed SS and whether that phosphorylation regulates downstream components that Liu et al have identified in this manuscript.

We thank the reviewer for raising this important point regarding PMK-1/p38 MAPK signaling. As established in our prior work (Reference 1), SS exposure triggers phosphorylation of PMK-1 (P-PMK-1) in *C. elegans*, and pmk-1 mutants exhibit enhanced growth on SS (Figure-1, Figure-2). This confirms that PMK-1-mediated innate immune signaling actively regulates SS responsiveness and digestion.

To address whether PMK-1 functions downstream of NSY-1 within our proposed model, we performed critical epistasis analyses. While we observed that nsy-1 mutation elevates ins-23 (indicating NSY-1 suppression of ins-23), knockdown of pmk-1 did not alter ins-23 expression levels (Figure 5-figure supplement 3C). This demonstrates that PMK-1 does not operate through the ins-23 pathway to regulate SS digestion. Thus, although both pathways respond to SS, the PMK-1-mediated innate immune response and the NSY-1/INS-23 axis constitute distinct regulatory mechanisms governing digestive adaptation.

Reference 1: Geng, S., Li, Q., Zhou, X., Zheng, J., Liu, H., Zeng, J., Yang, R., Fu, H., Hao, F., Feng, Q., & Qi, B. (2022). Gut commensal *E. coli* outer membrane proteins activate the host food digestive system through neural-immune communication. Cell host & microbe, 30(10), 1401–1416.e8. https://doi.org/10.1016/j.chom.2022.08.004

(2) Since p38 MAPK pathway has a well-established role in host defense in the *C. elegans* intestine, it is important to show that NSY-1 does not function in the intestine in the model that Liu et al present. I would like the authors to reintroduce nsy-1 in *C. elegans* intestine in nsy-1 mutant animals and then test whether it has any effect on worm length on SS food (similar to what is done in Figure 3 for AWC-specific nsy-1).

Beyond its established role in AWC neurons, we detected NSY-1 expression in the intestine (Figure 3-figure supplement 2A). To assess intestinal NSY-1 function, we performed tissue-specific rescue experiments in nsy-1 mutants using the intestinal-specific vha-1 promoter. Intestinal expression of NSY-1 significantly suppressed the enhanced SS digestion phenotype in nsy-1 mutants (Figure 3-figure supplement 2B), demonstrating functional involvement of gut-localized NSY-1 in regulating digestive responses. We propose intestinal NSY-1 mediates this effect through innate immune signaling, consistent with its known pathway components. As previously established (Reference 1), the canonical PMK-1/p38 MAPK pathway functions downstream of NSY-1, with both sek-1 and pmk-1 knockdown enhancing SS digestion through immune modulation. This indicates intestinal NSY-1 suppresses digestion may act through PMK-1-mediated immune responses. Since neuronal NSY-1's role in digestive control was previously undefined, we prioritized mechanistic analysis of its neuronal function in digestion regulation.

Notably, this immune-mediated mechanism operates independently of NSY-1's neuronal regulation pathway. In AWC neurons, NSY-1 controls digestion exclusively through the neuropeptide signaling axis (INS-23/DAF-2/BCF-1) without engaging innate immune components.

Reference 1: Geng, S., Li, Q., Zhou, X., Zheng, J., Liu, H., Zeng, J., Yang, R., Fu, H., Hao, F., Feng, Q., & Qi, B. (2022). Gut commensal *E. coli* outer membrane proteins activate the host food digestive system through neural-immune communication. Cell host & microbe, 30(10), 1401–1416.e8. https://doi.org/10.1016/j.chom.2022.08.004

(3) At multiple places, wild-type (WT) controls have been labeled as N2. It is better to label all controls as WT (and not as N2).

Corrected.

(4) In Figure 2B, the aversion response should be scored at multiple time points, like Figure 1C, rather than at just one timepoint.

We thank the reviewer for suggesting multi-timepoint analysis of aversion behavior. In accordance with this recommendation, we have now quantified SS avoidance at multi-timepoint. As shown in the revised Figure 2B, nsy-1 mutants exhibited significantly impaired avoidance responses at both 4h and 6h but not at 8h, confirming that NSY-1 is essential for sustained aversion to SS food in the early response. This data demonstrates that the critical role of NSY-1 in food discrimination at initial sensory responses.

(5) Does the re-introduction of nsy-1 in AWC neurons in nsy-1 mutant background help animals avoid SS in dwelling and food-choice assays? Along the same lines, does the CRISPR-generated AWC-specific mutant of NSY-1 fail to avoid SS in dwelling and food-choice assays similar to the whole-animal mutant? These behavioral data are missing in Figure 3.

We thank the reviewer for prompting behavioral validation of AWC-specific nsy-1 functions. To determine whether NSY-1 in AWC neurons mediates SS sensory perception, we performed dwelling (avoidance) and food-choice assays using AWC-specific nsy-1 knockout and AWC-rescued strains (nsy-1(ag3); Podr-1::nsy-1). In dwelling assays, AWC-specific nsy-1 KO mutants exhibited significantly impaired SS avoidance at 6h (Figure 3-figure supplement 3A), while AWC-rescued strains restored avoidance capacity at 2-6h (Figure 3-figure supplement 3B). Food-choice assays further revealed that AWC nsy-1 KO mutants preferentially migrated toward SS (Figure 3-figure supplement 3C), whereas AWC-rescued showed no preference between SS and HK-*E. coli* (Figure 3-figure supplement 3D). These data conclusively demonstrate that NSY-1 acts in AWC neurons to mediate SS recognition and aversion behaviors.

(6) In Figure 3E and F, the number of animals that were used for scoring AWC str-2p::GFP expression should be specified.

we added the number of animals in the figure.

(7) RNA seq analysis identified multiple GPCRs (including STR-130) that are upregulated in an NSY-1-dependent manner when animals are fed with SS bacteria. However, the authors decided to only characterize STR-130 because of previously published findings. It is important to rule out the role of other GPCRs since all are upregulated on SS food as shown in Figure S4 B. I would like the authors to knock down other GPCRs in the same manner as they did for STR-130 and demonstrate that only str-130 knockdown behaves similarly to the nsy-1 mutant (if that is the case) using the assay presented in Figure 4 D.

We appreciate the reviewer’s suggestion to comprehensively evaluate NSY-1-regulated GPCRs. In response, we extended our functional analysis to all six GPCRs (str-130, str-230, str-87, str-112, str-160, and sra-32) identified as NSY-1-dependent and SS-induced in RNA-seq (Figure 4—figure supplement 1).

Using RNAi knockdown and the SS growth assay, we observed that RNAi of str-130, str-230, str-87, or str-112 significantly enhanced SS growth (Figure 4—figure supplement 2A), with str-130 RNAi exhibiting the most robust phenotype—phenocopying nsy-1 mutants. Crucially, none of these GPCR knockdowns further enhanced growth in nsy-1(ag3) mutants (Figure 4—figure supplement 2B), confirming their position downstream of NSY-1. These data establish str-130 as the dominant effector of NSY-1-mediated SS response regulation, while suggesting minor contributions from other GPCRs (str-230, str-87, str-112).

(8) In Figure 4E and G, the number of animals that were used for scoring GFP expression should be specified.

we added the number of animals in the figure.

(9) When comparing Figure 3E and Figure 4E, it appears that the loss of str-130 RNAi does not phenocopy nsy-1 mutant. This raises the question of whether the inefficiency of RNAi targeting str-130 is the cause, or if STR-130 is not the only GPCR regulated by NSY-1 on SS food. I would like the authors to address this discrepancy. If RNAi inefficiency is indeed the cause, using an RNAi-sensitive background, such as an eri- 1 mutant, could help strengthen the data presented in Figure 4E. Conversely, if RNAi inefficiency is not responsible for the discrepancy, I suggest that the authors investigate the roles of other GPCRs that were identified by RNA sequencing.

We appreciate the reviewer’s observation regarding the phenotypic difference between nsy-1 mutants and str-130 (RNAi) animals on SS food (Fig. 3E vs Fig. 4E).

While both genetic perturbations significantly enhance SS growth and increase the proportion of animals exhibiting AWC^ON^ states compared to wild type (indicating enhanced digestion), the specific AWC^ON^ neuron configurations differ: nsy-1 mutants predominantly show 2 AWC^ON^ animals, whereas str-130(RNAi) animals primarily exhibit the 1 AWC^ON^ /1 AWC^OFF^ configuration (Fig. 3E vs Fig. 4E).

This difference likely arises because STR-130 is the key GPCR mediating NSY-1's inhibitory effect on SS digestion, but it is not the sole GPCR involved, as evidenced by our RNAi screen identifying several additional NSY-1-regulated GPCRs (str-230, str-87, str-112) whose depletion also enhanced SS growth (Fig. 4A-D).

The robust SS growth enhancement and AWC^ON^ state increase caused by str-130 (RNAi) (phenocopying the nsy-1 mutant’s functional outcome of enhanced digestion) (Figure 4D, 4E) indicate effective RNAi knockdown for this specific assay. Therefore, the distinct neural configurations reflect the partial redundancy among GPCRs downstream of NSY-1, rather than an inherent inefficiency of the str-130 RNAi.

The nsy-1 mutant phenotype represents the complete loss of all inhibitory GPCR signaling coordinated by NSY-1, while str-130(RNAi) represents the loss of its major component. Investigating the roles of other identified GPCRs (str-230, str-87, str-112) in modulating AWC^ON^ neuron states is an important direction for future research.

(10) In Figure 4 F and 4 G, the authors show that the overexpression of STR-130 rescues the nsy-1 mutant phenotype suggesting that NSY-1 might function through STR-130 to control digestion on SS food. These data place STR-130 downstream of NSY-1. To further strengthen these epistasis data, authors should knock down str-130 in nsy-1 mutant animals and show that the combined loss of both genes produces the same effect as the loss of either gene alone.

We thank the reviewer for the insightful suggestion to further define the genetic relationship between nsy-1 and str-130. To strengthen our epistasis analysis, we performed RNAi knockdown of str-130 in the nsy-1(ag3) mutant background and assessed development on SS food. Consistent with STR-130 acting downstream of NSY-1, the loss of str-130 via RNAi did not further enhance the developmental capacity (i.e., growth phenotype) of nsy-1(ag3) mutant animals on SS. This lack of enhancement indicates that str-130 and nsy-1 function within the same genetic pathway, with str-130 acting epistatically downstream of nsy-1 (Figure 4—figure supplement 3). This finding reinforces the model proposed from our overexpression data (Fig. 4F-G) – that NSY-1 primarily exerts its inhibitory effect on SS digestion by inducing the expression GPCR STR-130.

(11) In Figure 5C, please mention "ins-23 transcript levels" on the top of the graph so that it is clear what these data represent.

We appreciate the reviewer’s suggestion.

(12) Since all ins genes were upregulated in nsy-1 mutants (though ins-23 was indeed the most highly upregulated gene) on SS food from RNA seq analysis (Figure S5 B), it is important to first phenotypically characterize all of them using "worm length assay". If this analysis shows that ins-23 has the most robust phenotype, it would make more sense to just focus on ins-23.

We agree with the reviewer that initial phenotypic characterization of candidate genes identified through transcriptomic analysis is valuable.Our RNA-seq data revealed that several insulin-like peptide genes, including ins-22, ins-23, ins-24, and ins-27, were significantly upregulated in the nsy-1 mutant on SS food (Figure 5—figure supplement 1B). We prioritized these insulin-like peptide genes for functional validation because they are known to act as neuropeptides capable of mediating non-cell autonomous signaling in previous studies (Shao et al 2016).

To determine if any were functionally responsible for the enhanced SS growth observed in nsy-1 mutants, we performed functional phenotypic screening using the SS growth assay (worm length assay). We individually knocked down each of these candidates (ins-22, ins-23, ins-24, ins-27) in the nsy-1(ag3) mutant background. Among these, only RNAi targeting ins-23 significantly attenuated (i.e., suppressed) the enhanced development of the nsy-1(ag3) mutant on SS (Figure 5—figure supplement 2). This targeted functional screening revealed that ins-23 has the most robust and specific role in mediating the enhanced digestion phenotype downstream of NSY-1 loss, providing the critical justification for our subsequent focus on this particular insulin-like peptide.

Ref:

Shao, L. W., Niu, R., & Liu, Y. (2016). Neuropeptide signals cell non-autonomous mitochondrial unfolded protein response. Cell research, 26(11), 1182–1196. https://doi.org/10.1038/cr.2016.118

**Reviewer #2 (Recommendations for the authors):**
There are several minor errors and typos in the manuscript(1) A number of typos in the figures, like "length".

Corrected.

(2) The 'axis labels' are inconsistent from panel to panel, like "relative body length" and "relative worm length".

Corrected.

(3) The fonts are inconsistent from panel to panel.

Corrected.

(4) There is no Ex unique number for transgenic lines.

Corrected.

**Reviewer #3 (Recommendations for the authors):**

Minor points:

(1) Figure 3B, 3C, 3G, 4D, 4F, 5D, 5E, and 6C: Replace "lenth" with "length" (consistent with Figure 2A).

Corrected.

(2) Figure 4D: Correct "ctontrol" to "control."

Corrected.

(3) Figure 4G: Update the co-injection marker to Podr-1::GFP instead of Pstr-2::GFP.

Corrected.

(4) Figure 5C: This figure is missing from the Results section.

Corrected.

(5) Figure 6A: Label the graph with Pbcf-1::bcf-1::GFP, as in Figure 6D.

Corrected.

(6) Italicization: Lines 588 and 603-italicize nsy-1.

Corrected.

(7) Supplementary Figure S2A: Correct "Screeng" to "Screening."

Corrected.

(8) Spelling/Proofreading: Ensure consistent spelling and grammar, such as correcting "mutan" to "mutant" in Figure 4A.

Corrected.